# Kalman Filter Is All You Need: Optimization Works When Noise Estimation Fails

## Abstract

Determining the noise parameters of a Kalman Filter (KF) has been studied for decades. A huge body of research focuses on the task of noise estimation under various conditions, since precise noise estimation is considered equivalent to minimization of the filtering errors. However, we show that even a small violation of the KF assumptions can significantly modify the *effective* noise, breaking the equivalence between the tasks and making noise estimation an inferior strategy. We show that such violations are common, and are often not trivial to handle or even notice. Consequentially, we argue that a robust solution is needed – rather than choosing a dedicated model per problem. To that end, we apply gradient-based optimization to the filtering errors directly, with relation to an efficient parameterization of the symmetric and positive-definite parameters of the KF. In a variety of state-estimation and tracking problems, we show that the optimization improves both the accuracy of the KF and its robustness to design decisions. In addition, we demonstrate how an optimized neural network model can seem to reduce the errors significantly compared to a KF – and how this reduction vanishes once the KF is optimized similarly. This indicates how complicated models can be wrongly identified as superior to the KF, while in fact they were merely more optimized.

## 1 Introduction

The Kalman Filter (KF) (Kalman, 1960) is a celebrated method for linear filtering and prediction, with applications in many fields including tracking, guidance, navigation and control (Zarchan and Musoff, 2000; Kirubarajan, 2002). Due to its simplicity and robustness, it remains highly popular – with over 10,000 citations in the last 5 years alone (Google Scholar, 2021) – despite the rise of many non-linear sequential prediction models (e.g., recurrent neural networks). The KF relies on the following model for a dynamic system:

$$\begin{aligned} X_{t+1} &= F_t X_t + \omega_t & (\omega_t \sim N(0, Q)) \\ Z_t &= H_t X_t + \nu_t & (\nu_t \sim N(0, R)) \end{aligned} \tag{1}$$

where $X_t$ is the state of the system at time $t$ (whose estimation is usually the goal), and its dynamics are modeled by the linear operator $F_t$ up to the random noise $\omega_t$ with covariance $Q$; and $Z_t$ is the observation, which is modeled by the linear operator $H_t$ up to the noise $\nu_t$ with covariance $R$. When the operators $F_t, H_t$ are not assumed to depend on time, the notation may be simplified to $F, H$.

To use the KF, one must determine the noise parameters $Q, R$. The filtering errors (i.e., estimation errors of the states $\{X_t\}$) are minimized when $Q$ and $R$ correspond to the true covariance matrices of the noise (Humpherys et al., 2012). Thus, these parameters are usually determined by noise estimation. In absence of system state data $\{x_t\}$ (the "ground truth"), many methods have been suggested to determine $Q$ and $R$ from observations data $\{z_t\}$ alone (Abbeel et al., 2005; Odelson et al., 2006; Zanni et al., 2017; Park et al., 2019). When ground-truth data *is* available, however, noise estimation trivially reduces to calculation of the sample covariance matrices (Lacey, 1998):

$$\hat{R} := Cov(\{z_t - H_t x_t\}_t), \qquad \hat{Q} := Cov(\{x_{t+1} - F_t x_t\}_t). \tag{2}$$

Indeed, as stated by Odelson et al. (2006), "*the more systematic and preferable approach to determine the filter gain is to estimate the covariances from data*". Our work focuses on such problems with ground-truth available for learning (but not for inference after the learning, of course), which was motivated by a real-world Doppler radar estimation problem.

**Noise estimation is often not optimal:** The equivalence between noise estimation and errors minimization can be proved under the standard KF assumptions – that is, known and linear dynamics and observation models ($F_t, H_t$), with i.i.d and normally-distributed noises ($\{\omega_t\}, \{\nu_t\}$) (Humpherys et al., 2012). However, as put by Thomson (1994), "*experience with real-world data soon convinces one that stationarity and Gaussianity are fairy tales invented for the amusement of undergraduates*" – and linearity and independence can be safely added to this list. Therefore, under realistic assumptions, the covariance of the noise does not necessarily correspond to optimal filtering.

We introduce a case study in the context of radar tracking, where we demonstrate that even using the true covariance of the noise ("oracle" noise-estimation) is sub-optimal in a variety of scenarios – including very simple scenarios with relatively minor violation of the KF assumptions. In Appendices E and F, we also analyze this phenomenon analytically for two private cases (non-linearity in Doppler radar and non-i.i.d noise in lidar), where the violation of a single KF assumption is shown to modify the effective noise. By providing this extensive evidence that noise-estimation is a sub-optimal way to tune a KF even in presence of system-states ground-truth data, **we re-open a problem that was considered solved for decades** (Kalman, 1960).

We also show that seemingly small changes in the properties of the scenario may lead to major changes in the desired design of the KF, e.g., whether to use a KF or an Extended KF (Sorenson, 1985). In certain cases, the design choices are easy to overlook (e.g., Cartesian vs. spherical coordinates), and are not trivial to make even if noticed. As a result, it is impractical to manually choose or develop a variant of the KF for every problem. Rather, we should assume that our model is sub-optimal, and leverage data to deal with the sub-optimality as robustly as possible.

**Optimization *is* optimal:** We consider $Q$ and $R$ as model parameters that should be optimized with respect to the filtering errors (i.e., system-state estimation errors) – rather than estimating the noise. While both noise estimation and errors optimization rely on exploitation of data, only the latter explicitly addresses the actual goal of solving the filtering problem.

Gradient-based optimization methods are usually effective in the field of machine learning, but applying them naively to the entries of $Q$ and $R$ may violate the symmetry and positive-definiteness (SPD) constraints of the covariance matrices. Indeed, even works that come as far as optimizing $Q$ and $R$ (instead of estimating the noise) usually apply limited optimization methods, e.g., grid-search (Coskun et al., 2017) or diagonal restriction of the covariance matrices (Formentin and Bittanti, 2014; Li et al., 2019). To address this issue, we use a parameterization based on Cholesky decomposition (Horn and Johnson, 1985), which allows us to apply gradient-based optimization to SPD matrices. This method is computationally efficient compared to other general gradient-based methods for SPD optimization (Tsuda et al., 2005; Tibshirani, 2015).

We demonstrate that the optimization reduces the errors of the KF consistently: over different variants of the KF, over different violations of KF assumptions, over different domains (tracking from radar, video or lidar), over small and large training datasets, and even under distributional shifts between train and test datasets. Furthermore, we show that optimization improves the robustness to design decisions, by shrinking the performance gaps between different variants of the KF.

As explained above, we extensively justify the need to optimize the KF in many practical problems, and suggest a simple solution which is effective, robust, computationally efficient, and relies on standard tools in supervised machine learning. As a result, we believe that in the scope of filtering problems with available ground-truth, whenever the KF assumptions are not strictly-guaranteed, **the suggested optimization method should become the new standard procedure for the KF tuning**.

**Unfair comparison:** Many learning algorithms have been suggested to address non-linearity in filtering problems, e.g., based on Recurrent Neural Networks (RNN). Such works often use a linear tool such as the KF as a baseline for comparison – with tuning parameters being sometimes ignored (Gao et al., 2019), sometimes based on noise estimation (fa Dai et al., 2020), and sometimes optimized in a limited manner using trial-and-error (Jamil et al., 2020) or grid-search (Coskun et al., 2017). Our findings imply that such a methodology yields over-optimistic conclusions, since the baseline is not optimized to the same level as the learning model. This may result in adoption of over-complicated algorithms with no actual added value. Instead, any learning algorithm should be compared to a baseline that is optimized using a similar method (e.g., gradient-descent with respect to the errors).

Indeed, we consider an extension of the KF based on LSTM, which is the key component in many SOTA algorithms for non-linear sequential prediction in recent years (Neu et al., 2021). For radar tracking with non-linear motion, we demonstrate how the LSTM seems to provide a significant improvement over the KF. Then, we show that the whole improvement comes from optimization of parameters, and *not* from the expressive non-linear architecture. In particular, this result demonstrates the competitiveness of our suggested method versus SOTA sequential prediction models.

Recent works in the area of machine learning have already shown that advanced algorithms often obtain most of their improvement from implementation nuances (Engstrom et al., 2019; Andrychowicz et al., 2020; Henderson et al., 2017). Our work continues this line of thinking and raises awareness to this issue in the domain of filtering problems.

**Contribution:**  We show (empirically and analytically) that the gold-standard noise estimation method to tune the KF given ground-truth data is often sub-optimal; demonstrate how this leads to underevaluation of the KF compared to optimized filtering models such as neural networks; suggest a simple method to optimize the KF, using gradient-based optimization with Cholesky parameterization; and extensively demonstrate its improved accuracy and its robustness to model misspecification.

**Limitations:**  This work relies on the availability of ground-truth system-states in the training data to allow supervised learning. Ground-truth data is often available from simulations, controlled experiments or manual labeling. Within this scope, we show that although noise estimation is straight-forward (Equation 2), it is often not the right task to address. Another limitation is in the theoretical guarantees of gradient-based optimization (see Appendix L), despite its wide success in many fields.

The paper is organized as follows: Section 2 reviews the KF. Section 3 introduces our method for efficient KF optimization. Section 4 justifies the necessity of KF optimization through a detailed case study. Section 5 presents a neural version of the KF which reduces the errors compared to a standard KF – but not when compared to an optimized KF. Section 6 discusses related works.

## 2   PRELIMINARIES: THE KALMAN FILTER ALGORITHM

The KF algorithm (Kalman, 1960; Humpherys et al., 2012) relies on Equation 1 for a dynamic system model. It keeps an estimate of the state $X_t$, represented as the mean $x_t$ and covariance $P_t$ of a normal distribution. As shown in Figure 1, it alternately predicts the next state using the dynamics model (*prediction* step), and processes new information from incoming observations (*update* or *filtering* step).

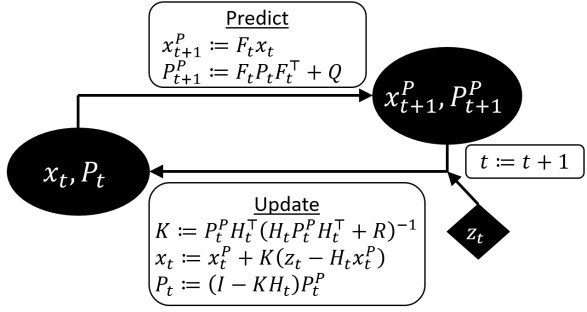

Figure 1: The KF algorithm. The prediction step is based on the motion model $F_t$ with noise $Q$, whereas the update step is based on the observation model $H_t$ with noise $R$.

The KF yields optimal state estimations – but only under a restrictive set of assumptions (Kalman, 1960), as specified in Definition 2.1. Note that normality of the noise is excluded since it is not necessary for optimality (Humpherys et al., 2012), although it is also often assumed.

**Definition 2.1** (KF assumptions). $F_t, H_t$ of Equation 1 are known matrices that do not depend on the state (i.e., correspond to linear models); $\omega_t, \nu_t$ are i.i.d random variables with zero-mean and constant, known covariance matrices $Q, R$, respectively; and the initial state distribution is known.

Certain assumptions violating in Definition 2.1 can be partially handled by variations of the KF, such as the Extended KF (EKF) (Sorenson, 1985) which replaces the linear models $F_t, H_t$ with local linear approximations, and the Unscented KF (UKF) (Wan and Van Der Merwe, 2000) which applies the filtering through sigma-points sampled from the estimated distribution. The use of multiple tracking models alternately is also possible using switching mechanisms (Mazor et al., 1998).

While $F_t$ and $H_t$ are usually determined based on domain knowledge, $Q$ and $R$ are often estimated from data as the covariance of the noise. As mentioned in Section 1, this can be done using Equation 2

if ground-truth data is available, or using more sophisticated methods otherwise (Odelson et al., 2006; Feng et al., 2014; Park et al., 2019).

See Appendix A for a detailed introduction of the KF and recurrent neural networks (RNN, LSTM).

## 3 KF OPTIMIZATION USING CHOLESKY PARAMETERIZATION

The performance of a KF strongly depends on its parameters $Q$ and $R$ (Formentin and Bittanti, 2014). These parameters are usually regarded as estimators of the noise covariance in motion and observation, respectively (Lacey, 1998), and are estimated accordingly. Although optimization has been suggested for the KF in the past (Abbeel et al., 2005), it was often viewed as a fallback solution (Odelson et al., 2006), for cases where direct estimation is not possible (e.g., the true states are unavailable in the data (Feng et al., 2014)). Accordingly, we define our baseline method for this work:

**Method 1** (Estimated KF). KF whose parameters $Q, R$ were determined from data using Equation 2.

The preference of noise estimation relies on the fact that the KF – with parameters corresponding to the noise covariances – minimizes the square filtering errors (MSE) of the estimates of the system-states $\{X_t\}$. This holds under Assumptions 2.1 (Humpherys et al., 2012). Hence, equivalently, we could explicitly look for the parameters that minimize the MSE, e.g., using the Adam algorithm (Diederik P. Kingma, 2014). Adam is a popular variant of the well-known gradient-descent algorithm, and has achieved remarkable results in many optimization problems in the field of machine learning in recent years, including in non-convex problems where local-minima exist (Zhong et al., 2020).

Numeric optimization is more complicated than computing two covariance matrices, but is shown to be beneficial in Sections 4 and 5. Stable optimization algorithms are available in several open-source packages, e.g., PyTorch (Paszke et al., 2019), which supports gradient-propagation through matrix-inversion (as needed for the KF computations). As demonstrated in Sections 4,5, the PyTorch implementation of Adam provides stable results on MSE-optimization of a KF. This is conceivable, as the KF is arguably a simple model in terms of mathematical sophistication and number of parameters.

One major challenge in the KF parameters optimization is that both $Q$ and $R$ correspond to covariance matrices, which are symmetric and positive definite (SPD): a naive numeric optimization of their entries may ruin the SPD structure. This difficulty often motivates optimization methods that avoid gradients (Abbeel et al., 2005), or even the restriction of $Q$ and $R$ to be diagonal (Li et al., 2019). Indeed, Formentin and Bittanti (2014) pointed out that "*since both the covariance matrices must be constrained to be positive semi-definite, $Q$ and $R$ are often parameterized as diagonal matrices*".

To allow Adam to optimize the non-diagonal $Q$ and $R$ we use the Cholesky parameterization (Pinheiro and Bates, 1996). It relies on Cholesky decomposition (Horn and Johnson, 1985), which states that any SPD matrix $A \in \mathbb{R}^{n \times n}$ can be written as $A = LL^\top$, where $L$ is lower-triangular with positive entries along its diagonal. The reversed claim is also true: for any lower-triangular $L$ with positive diagonal, $LL^\top$ is SPD. Now consider the parameterization of $A$ using the following $\binom{n}{2} + n = \frac{n(n+1)}{2}$ parameters: $\binom{n}{2}$ parameters correspond to $\{L_{ij}\}_{1 \leq j < i \leq n}$, and $n$ parameters to $\{\log L_{ii}\}_{1 \leq i \leq n}$. Clearly, the transformation from the parameters to the SPD matrix $A = LL^\top$ is differentiable and outputs a SPD matrix for any realization of the parameters in $\mathbb{R}^{n(n+1)/2}$. Thus, we can apply Adam (or any other gradient-based method) to optimize these parameters without worrying about the SPD constraints. This approach, which is presented in a more general framework in Appendix C and is implemented in our code, is used in our KF optimization method as follows:

**Method 2** (Optimized KF). A KF whose parameters $Q, R$ were determined from data by optimizing the $MSE$ of the state-estimates, using Adam algorithm with relation to a Cholesky parameterization of $Q$ and $R$, as described in detail in Appendix B.1.

Note that other methods exist for SPD optimization, but require SVD-decomposition every iteration and thus are computationally heavier, e.g., matrix-exponent (Tsuda et al., 2005) and projected gradient-descent with respect to the SPD cone (Tibshirani, 2015). The Cholesky parameterization only requires a single matrix multiplication, and thus is both efficient and easy to implement. To the best of our knowledge, this parameterization is not used at all for supervised optimization of KF.

## 4 KALMAN FILTER CONFIGURATION AND OPTIMIZATION: A CASE STUDY

In this section, we introduce a detailed case study to compare noise estimation and errors optimization in the KF. As mentioned in Section 2, the two are equivalent only under Assumptions 2.1 (Humpherys et al., 2012). Some of these assumptions are clearly violated in realistic scenarios, while other violations may be less obvious. For example, even if a radar's noise is i.i.d in spherical coordinates, it is not so in Cartesian coordinates (see Appendix A.5).

The need to rely on many assumptions might explain why there are several extensions and design decisions in a KF configuration. This includes the choice between KF and EKF; the choice between educated state initialization and a simple uniform prior (Linderoth et al., 2011); and certain choices that may be made without even noticing, such as the coordinates of the state representation.

The case study below justifies the following claims:

1. Design decisions in a KF are often nontrivial to make and are potentially significant.
2. Tuning a KF by noise estimation is often highly sub-optimal – even in very simple scenarios.
3. Tuning a KF by optimization improves both accuracy and robustness to design decisions.
4. KF optimization using Method 2 is robust to distributional shifts (Appendix G) and to small training datasets (Appendix H).

These claims imply that the popular KF algorithm may not be exploited to its full potential. Furthermore, many works that compare learning algorithms to a KF baseline conduct an "unfair" comparison, as the learning algorithms are optimized and the KF is not. This may lead to adoption of unnecessarily complicated algorithms, as demonstrated in Section 5. Indeed, in many works the tuning of the baseline KF is either ignored in the report (Gao et al., 2019) or relies on estimation (or knowledge) of the noise (fa Dai et al., 2020; Jamil et al., 2020), as demonstrated extensively in Section 6.

### 4.1 SETUP AND METHODOLOGY

In the case study of **radar tracking**, each target is represented by a sequence of (unknown) states in consecutive time-steps, and a corresponding sequence of radar measurements. A state $x_{full} = (x_x, x_y, x_z, x_{vx}, x_{vy}, x_{vz})^\top \in \mathbb{R}^6$ consists of 3D location and velocity. We also denote $x = (x_x, x_y, x_z)^\top \in \mathbb{R}^3$ for the location alone (or $x_{\text{target},t} \in \mathbb{R}^3$ for the location of a certain target at time $t$). An observation $z \in \mathbb{R}^4$ consists of noisy measurements of range, azimuth, elevation and Doppler signal. Note that the former three correspond to a noisy measurement of $x$ in spherical coordinates, and the latter one measures the projection of velocity onto the radial direction $x$. The goal is to minimize the error of the point-estimate of the state: $MSE = \sum_{\text{target}} \sum_t (\tilde{x}_{\text{target},t} - x_{\text{target},t})^2$.

The case study considers 5 types of tracking scenarios (*benchmarks*) and 4 variants of the KF (*baselines*). For each benchmark and each baseline, we use the benchmark training data to produce one estimated **KF** (Method 1) and one optimized KF (**OKF**, Method 2). We then evaluate the errors of both models on the test data of the benchmark (generated using different seeds than the training data). All experiments were run on eight i9-10900X CPU cores on a single Ubuntu machine.

Figures 2b,2c display a sample of trajectories in the simplest benchmark (*Toy*), which satisfies all KF assumptions except for a linear observation model $H$; and in the complex *Free Motion* benchmark, which violates several assumptions. The other benchmarks are demonstrated in Appendix M. Figure 2a defines each of the 5 benchmarks more formally as a certain subset of the following properties:

- *anisotropic*: horizontal motion is more likely than vertical motion (otherwise motion direction is distributed uniformly).
- *polar*: radar noise is generated i.i.d in spherical coordinates (otherwise noise is Cartesian i.i.d, which violates the physics of the system).
- *uncentered*: targets are not forced to concentrate close to the radar.
- *acceleration*: speed change is allowed (through intervals of constant acceleration).
- *turns*: non-straight motion is allowed.

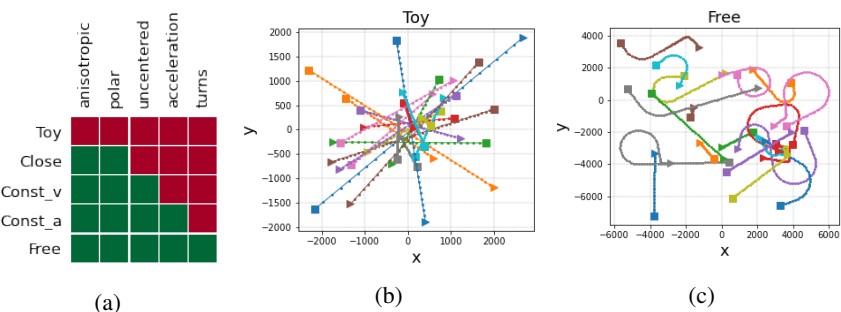

(a)        (b)        (c)

Figure 2: (a) Benchmarks names (rows) and the properties that define them (columns). Green means that the benchmark satisfies the property. (b,c) Targets in Toy and Free Motion benchmarks (projected onto XY plane).

The 4 baselines differ from each other by using either KF or EKF, with either Cartesian or spherical coordinates for representation of $R$ (the rest of the system is always represented in Cartesian coordinates). As mentioned above, each baseline is tuned once by Method 1 and once by Method 2. Note that the non-linear observation model requires minor adjustments of the KF baselines and the tuning methods, as explained in Appendix A.3. For the Toy benchmark, the optimal parameters are also derived **analytically** in Appendix E. Appendix G demonstrates the robustness of OKF to distributional shifts by training on one benchmark and testing on another. Appendix H repeats the tests for different sizes of training datasets.

We also repeat the experiment for the problem of **tracking from video**, using MOT20 dataset (Dendorfer et al., 2020) of real-world pedestrians, with train and test datasets taken from separated videos. For this problem we only consider a KF with Cartesian coordinates, since there is no polar component in the problem. See Appendix I for the detailed setup of the video tracking experiments. In addition, we test the OKF in the problem of **self-driving state-estimation from lidar measurements** (Appendix J). These two domains extend the scope of the experimented KF assumptions violations, since they correspond to linear observation models (unlike the radar benchmarks), with both noisy (lidar) and noiseless (MOT20) observations. For a simplified lidar problem, we also derive the optimal parameters **analytically** in Appendix F.

## 4.2 RESULTS

**Design decisions are not trivial:** Table 1 summarizes the tracking errors. The left column in each cell corresponds to Method 1 (standard KF), and shows that in each benchmark, the errors strongly depend on the design decisions ($R$'s coordinates and whether to use EKF). In the Toy benchmark, for example, EKF is the best design, since the observation model $H$ is non-linear.

In other benchmarks, however, the winning designs of the non-optimized KF are arguably surprising:

1. Under non-isotropic motion direction (all benchmarks except Toy), EKF is worse than KF despite the non-linear motion. It is possible that since the horizontal prior reduces the stochasticity of $H$, the advantage of EKF no longer justifies the instability of the derivative-based approximation.

Table 1: Summary of the errors of the various models over the various benchmarks (on out-of-sample test data). Corresponding confidence intervals are available in Figure 14a in the appendix. In the model names, "O" denotes optimized, "E" denotes extended, and "p" denotes polar, For example, OEKFp is an *extended* KF with *polar* (spherical) representation of $R$ and *optimized* parameters. For KFp, we also consider an oracle-realization of $R$ according to the true noise of the simulated radar in spherical coordinates (available only in spherical benchmarks). Note that (1) for any benchmark and any baseline, optimization yields lower errors than estimation; and (2) this remains true even in the oracle variant, where the noise "estimation" is perfect.

| Benchmark | KF | OKF | KFp | KFp (oracle) | OKFp | EKF | OEKF | EKFp | OEKFp |
|---|---|---|---|---|---|---|---|---|---|
| Toy | 151.7 | 84.2 | 269.6 | – | 116.4 | 92.8 | **79.4** | 123.0 | 109.1 |
| Close | 25.0 | 24.8 | 22.6 | 22.5 | **22.5** | 26.4 | 26.1 | 24.5 | 24.1 |
| Const_v | 90.2 | 90.0 | 102.3 | 102.3 | **89.2** | 102.5 | 99.7 | 112.7 | 102.1 |
| Const_a | 107.5 | 101.6 | 118.4 | 118.3 | **100.3** | 110.0 | 107.0 | 126.0 | 108.7 |
| Free | 125.9 | 118.8 | 145.6 | 139.3 | **117.9** | 135.8 | 121.9 | 149.3 | 120.0 |

2. Even when the observation noise is spherical i.i.d, spherical representation of $R$ is not beneficial unless the targets are forced to concentrate close to the radar (last 3 benchmarks in Table 1). It is possible that when the targets are distant, Cartesian coordinates have a more important role in expressing the horizontal prior of the motion.

Since the best variant of the KF for each benchmark seems difficult to predict in advance, a practical system cannot rely on choosing the KF variant optimally, and rather has to be robust to this choice.

**Optimization is more accurate and robust:** Table 1 shows that for *every* benchmark and *every* baseline in the **radar tracking** problem (20 experiments in total), optimization yielded smaller errors than noise estimation (over an out-of-sample test dataset). Note that OKF wins even in the Toy scenario, under the slightest violation of KF assumptions. In addition, the variance between the baselines reduces under optimization, i.e., the optimization makes the KF more robust to design decisions (which is also evident in Figure 14a in Appendix M).

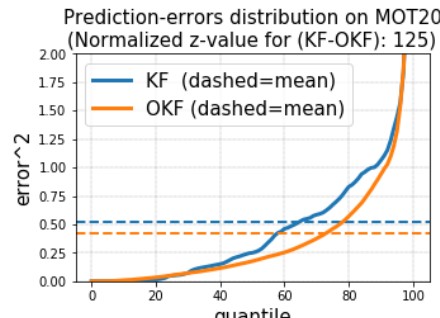

We also studied the performance of a KF with a perfect knowledge of the noise covariance matrix $R$. Note that in the constant-speed benchmarks, the estimation of $Q = 0$ is already very accurate, hence in these benchmarks the oracle-KF has a practically perfect knowledge of both noise covariances. Nonetheless, Table 1 shows that

Figure 3: Prediction errors of KF and OKF on 1208 targets of the test data of MOT20 videos dataset. The MSE of OKF is smaller by 18%, with statistical significance of p-value $< 10^{-6}$ over the test samples.

the oracle yields very similar results to a KF with estimated parameters. This indicates that the limitation of noise estimation in the KF indeed comes from choosing the wrong goal, and not from estimation inaccuracy.

Figure 3 shows that OKF provides significantly better predictions also in **video tracking**, where the MSE is reduced by 18% (see Appendix I for more details). In addition, in **lidar-based state estimation**, Appendix J shows a similar error reduction of 15%, and Lemma F.3 explains the $MSE$-advantage of OKF **analytically**.

|  | [Toy] KF / R | | | | | | [Toy] OKF / R | | | |
|  | x | y | z | Dop | | | x | y | z | Dop |
|---|---|---|---|---|---|---|---|---|---|---|
| x | 10048 | 59 | 14 | -0 | | x | 3231 | 13 | 1 | 8 |
| y | 59 | 10062 | 96 | -0 | | y | 13 | 2906 | 48 | -9 |
| z | 14 | 96 | 9959 | 2 | | z | 1 | 48 | 3087 | 3 |
| Dop | -0 | -0 | 2 | 25 | | Dop | 8 | -9 | 3 | 99 |

(a) Estimated $R$      (b) Optimized $R$

OKF is further demonstrated to be robust to major **distributional shifts** in Appendix G, and to **small training datasets** in Appendix H.

**Diagnosis of the KF sub-optimality in Toy scenario:** The source of the gap between estimated and optimized noise parameters can be studied through the simplest Toy benchmark, where the only violation of KF assumptions is the non-linear observation model $H$. Since the

Figure 4: The covariance matrix $R$ of the observation noise obtained in a (Cartesian) KF by noise estimation and by optimization, based on the dataset of the Toy benchmark. The axes correspond to the observation variables associated with the matrix entries. Note that the noise estimation is quite accurate, as the true variance of the noise is $100^2$ for the positional dimensions and $5^2$ for Doppler. The optimization increases the variance associated with the Doppler signal, as predicted by Lemma E.1. The decrease in the other diagonal components is discussed in Appendix E.

non-constant entries of $H$ correspond to the Doppler observation, the non-linearity inserts uncertainty to the Doppler observation (in addition to the inherent observation noise). This increases Doppler's effective noise in comparison to the location observation, as shown **analytically** by Lemma E.1 in the appendix. This explanation is consistent with Figure 4: the noise associated with Doppler is indeed increased by the optimization. Note that the non-linearity modifies the effective noise in a delicate way, which would *not* be compensated by a naive trial and error of noise inflation or deflation.

## 5 NEURAL KALMAN FILTER: IS NON-LINEAR PREDICTION HELPFUL?

A standard Kalman Filter for a tracking task assumes linear motion, as discussed in Section 2. In this section we introduce the *Neural Kalman Filter* tracking model (NKF), which uses LSTM networks to model non-linear motion, while keeping the framework of the KF – namely, the probabilistic representation of the target's state, and the separation between prediction and update steps.

Every prediction step, NKF uses a neural network model to predict the target's acceleration $a_t$ and the motion uncertainty $Q_t$. Every update step, another network predicts the observation uncertainty $R_t$. The predicted quantities are incorporated into the standard update equations of the KF, as shown in detail in Figure 7 in Appendix B. For example, the prediction step of NKF is:

$$x_{t+1}^P = Fx_t + 0.5a_t(\Delta t)^2, \qquad P_{t+1}^P = FP_tF^\top + Q_t$$

where $F = F(\Delta t)$ is the constant-velocity motion operator, and $a_t, Q_t$ are predicted by the LSTM (whose input includes the recent observation and the current estimated target's state). Other predictive features were also attempted but failed to provide significant value, as described in Appendix B. Note that the neural network predicts the acceleration rather than directly predicting the location. This is intended to regularize the predictive model and to exploit the domain knowledge of kinematics.

After training NKF over a dataset of simulated targets with random turns and accelerations (see Appendix B.1 for more details), we tested it on a test dataset. The test dataset is similar to the training dataset, with different seeds and an extended range of permitted accelerations. As shown in Figure 5, NKF significantly reduces the tracking errors compared to a standard KF.

At this point, it seems that the non-linear architecture of NKF provides better accuracy in the non-linear problem of radar tracking. However, Figure 5 shows that by shifting the baseline from a naive KF to an optimized one, we *completely eliminate* the advantage of NKF, and in fact reduce the errors even further. In other words, in this experiment the benefits of NKF come *only* from optimization and *not at all* from the expressive architecture. By overlooking the sub-optimality of noise estimation in the KF, we would wrongly adopt the over-complicated NKF.

Note that the optimized KF (OKF) also generalizes well to targets with different accelerations than observed in the training, which indicates a certain robustness to distribu-

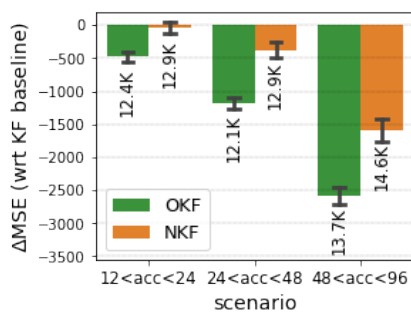

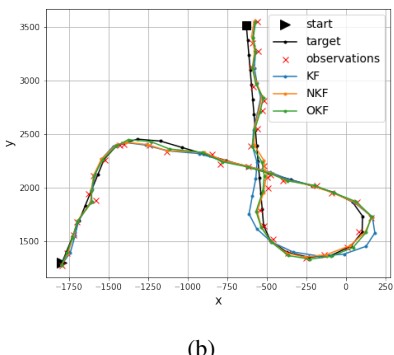

Figure 5: (a) Relative tracking errors (lower is better) with relation to a standard KF, over targets with different ranges of acceleration. The error-bars represent confidence intervals of 95%. The label of each bar represents the corresponding *absolute* MSE ($\times 10^3$). In the training data, the acceleration was limited to 24-48, hence the other ranges measure generalization. While the Neural KF (NKF) is significantly better than the standard KF, its advantage is entirely eliminated once we optimize the KF (OKF). (b) A sample target and the corresponding models outputs (projected onto XY plane). The standard KF has a difficulty to handle some of the turns.

tional shifts. Appendix K extends the experiment to additional variants of NKF, to another tracking benchmark, and to the evaluation metric of likelihood (NLL) in addition to estimation error (MSE). Note that high likelihood score is important for the matching task in a multi-target tracking problem.

Of course, our results do not imply that neural-networks in general cannot be superior to a KF: only that when comparing the two, if the KF is not optimized similarly to the neural model, the experimental results may be very misleading. As discussed in Section 6, this wrong *methodology* is not uncommon in the literature.

## 6    RELATED WORK

**Noise estimation:**    When tuning a KF, the system-states are often unavailable in the data (Formentin and Bittanti, 2014). Thus, estimation of the KF noise parameters from observations alone has been studied for decades, addressed using various methods based on autocorrelation (Mehra, 1970; Carew and Belanger, 1973), EM (Shumway and Stoffer, 2005) and others (Odelson et al., 2006; Feng et al., 2014; Park et al., 2019). However, if the states *are* available, noise estimation reduces to Equation 2 and is considered a solved problem (Odelson et al., 2006). In this case, we show that although noise estimation is easy, it is often not the right task to address.

Many works addressed the problem of non-stationary noise estimation (Zanni et al., 2017; Akhlaghi et al., 2017). However, as demonstrated in Sections 4,5, in certain cases stationary methods are highly competitive if tuned correctly – even in problems with complicated dynamics.

**Optimization:** In this work we apply gradient-based optimization to the KF with respect to its errors. Optimization without gradients was already suggested in Abbeel et al. (2005). In practice, "optimization" of the KF is often handled manually using trial-and-error (Jamil et al., 2020) or a grid-search over possible values of $Q$ and $R$ (Formentin and Bittanti, 2014; Coskun et al., 2017). In other cases, $Q$ and $R$ are restricted to be diagonal (Li et al., 2019; Formentin and Bittanti, 2014).

Gradient-based optimization of SPD matrices in general was suggested in Tsuda et al. (2005) using matrix-exponents, and is also possible using projected gradient-descent (Tibshirani, 2015) – both rely on SVD-decomposition. In this work, we apply gradient-based optimization using the parameterization that was suggested in Pinheiro and Bates (1996), which requires a mere matrix multiplication, and thus is both efficient and easy to implement.

In absence of a trajectories dataset, a recent line of works (Tsiamis and Pappas, 2020; Goel and Hassibi, 2021) applies online optimization within the current trajectory to minimize a regret metric.

**Neural Networks (NNs) in filtering problems:** Section 5 presents a RNN-based extension of the KF, and demonstrates how its advantage over the linear KF vanishes once the KF is optimized. The use of NNs for non-linear filtering problems is very common in the literature, e.g., in online tracking prediction (Gao et al., 2019; Dan Iter, 2016; Coskun et al., 2017; fa Dai et al., 2020; Ullah et al., 2019), near-online prediction (Kim et al., 2018), and offline prediction (Liu et al., 2019b). In addition, while Bewley et al. (2016) apply a KF for video tracking from mere object detections, Wojke et al. (2017) add to the same system a NN that generates visual features as well. NNs were also considered for related problems such as data association (Liu et al., 2019a), model-switching (Deng et al., 2020), and sensors fusion (Sengupta et al., 2019).

In many works that consider NNs for filtering problems, a KF is used as a baseline for comparison. However, while the NN parameters are typically optimized with respect to the filtering errors, the KF parameters tuning is sometimes ignored (Gao et al., 2019; Bai et al., 2020; Zheng et al., 2019), sometimes based on estimation (or knowledge) of the noise (fa Dai et al., 2020; Aydogmus and Aydogmus, 2015; Revach et al., 2021), and sometimes optimized in a limited manner as mentioned above (Jamil et al., 2020; Coskun et al., 2017; Ullah et al., 2019). Our findings imply that this methodology is wrong, since the baseline is not optimized to the same level as the learning model. Hussein (2014) explicitly discusses the sensitivity of EKF performance to the noise model accuracy, and suggests the solution of a NN with supervised learning – without considering the same supervised learning for the EKF.

## 7 SUMMARY

Through a detailed case study, we demonstrated both analytically and empirically the fragility of the KF assumptions, and how the slightest violation of them may change the effective noise in the problem – leading to significant and non-trivial changes in the optimal noise parameters. We addressed this problem using optimization tools from supervised machine learning, and suggested how to apply them efficiently to the SPD parameters of the KF.

We demonstrated the accuracy of our method over different violations of KF assumptions, in different domains (radar tracking, video tracking and lidar-based state estimation), with relation to different variants of the KF, over small and large training datasets, and even under distributional shifts between train and test datasets. Indeed, once we acknowledged the need for optimization and applied the Cholesky parameterization, the optimization itself performed robustly over all these scenarios. In light of this evidence, we recommend to use our method as the default procedure for the KF tuning in presence of ground-truth data, whenever the KF assumptions are not strictly-guaranteed.

We also demonstrated one of the consequences of using a sub-optimal KF: the common methodology of comparing learning filtering algorithms to classic variants of the KF is misleading, as it essentially compares an optimized model to a non-optimized one. We argued that the baseline method should be optimized similarly to the researched one, e.g., using optimization rather than noise estimation.

## REPRODUCIBILITY

All the experiments in this work are reproducible using our underline{code}, including data generation, models training and results analysis. The complete proofs for the theoretical results are available in Appendices E and F.

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

CONTENTS

# A PRELIMINARIES: EXTENDED

## A.1 MULTI-TARGET RADAR TRACKING

In the problem of multi-target radar tracking, noisy observations (or *detections*) of aerial objects (or *targets*) are received by the radar and are used to keep track of the objects. In standard radar sensors, the signal of an observation includes the target range and direction (which can be converted into location coordinates $x, y, z$), as well as the Doppler shift (which can be converted into the projection of velocity $v_x, v_y, v_z$ onto the location vector).

The problem consists of observations-trackers assignment, where each detection of a target should be assigned to the tracker representing the same target; and of location estimation, where the location of each target is estimated at any point of time according to the observations (Chang and Dunn, 2019). In the fully online setup of this problem, the assignment of observations is done once they are received, before knowing the future observations on the targets (though observations of different targets may be received simultaneously); and the location estimation at any time must depend only on previously received observations.

**Assignment problem:** The assignment problem can be seen as a one-to-one matching problem in the bipartite graph of existing trackers and newly-received observations, where the edge between a tracker and an observation represents "how likely the observation $z_j$ is for the tracker $trk_i$". In particular, if the negative-log-likelihood is used for edges weights, then the total cost of the match represents the total likelihood (under the assumption of observations independence):

$$C(\text{match}) = \sum_{(i,j)\in\text{match}} -\log P(z_j|trk_i) = -\log \prod_{(i,j)\in\text{match}} P(z_j|trk_i) \quad (3)$$

The assignment problem can be solved using standard polynomial-time algorithms such as the Hungarian algorithm (Kuhn, 1955). However, the assignment can only be as good as the likelihood information fed to the algorithm. This is a major motivation for trackers that maintain probabilistic representation of the target, rather than a merely point estimate of the location. A common example for such a tracking mechanism is the Kalman filter discussed below.

## A.2 KALMAN FILTER

Kalman Filter (KF) is a widely-used method for linear filtering and prediction originated in the 1960s (Kalman, 1960), with applications in many fields (Zarchan and Musoff, 2000) including object tracking (Kirubarajan, 2002). The classic model keeps an estimate of the monitored system state (e.g., location and velocity), represented as the mean $x$ and covariance $P$ of a normal distribution (which uniquely determine the PDF of the distribution). The mechanism (see Figure 1) alternately applies a *prediction step*, where a linear model $x := Fx$ predicts the next state; and an *update step* (also termed *filtering step*), where the information of a new observation $z$ is incorporated into the estimation (after a transformation $H$ from the observation space to the state space). In general, both models $F_t, H_t$ can be time-dependent, but in practice they are often time-invariant, thus the notation is often simplified to $F$ and $H$.

Note that KF compactly keeps our knowledge about the monitored target at any point of time, which allows us to estimate both whether a new observation corresponds to the currently monitored target, and what the state of the system currently is. However, KF strongly relies on several assumptions:

1. **Linearity**: both the state-transition of the target $f_t$ and the state-observation transformation $h_t$ are assumed to be linear, namely, $f_t(x) = F_t \cdot x$ and $h_t(x) = H_t \cdot x$. Note that the Extended KF described below, while not assuming linearity, still assumes known models for transition and observation.

2. **Normality**: both state-transition and state-observation are assumed to have a Gaussian noise with covariance matrices $Q$ and $R$ respectively. As a result, the estimates of the state $x$ and its uncertainty $P$ also correspond to the mean and covariance matrix of a Normal distribution representing the information regarding the target location.

3. **Known model**: $F_t, Q, H_t, R$ are assumed to be known.

While $F_t$ and $H_t$ are usually determined manually according to domain knowledge, the noise model parameters $R, Q$ are often estimated from data (as long as the the true states are available in the data). Specifically, they are often estimated from the sample covariance matrix of the noise: $R := Cov(Z - H_t X), Q := Cov(\Delta X) = Cov(\{x_{t+1} - F_t x_t\}_t)$ (Lacey, 1998).

Two non-linear extensions of KF – *Extended Kalman Filter* (EKF) (Sorenson, 1985) and *Unscented Kalman Filter* (UKF) (Wan and Van Der Merwe, 2000) – are also very popular in problems of non-linear state estimation and navigation (Wan, 2006). EKF replaces the linear prediction ($F_t$) and observation ($H_t$) models with non-linear known models $f_t$ and $h_t$, and essentially runs a standard KF with the local linear approximations $F_t = \frac{\partial f_t}{\partial x}|_{x_t}, H_t = \frac{\partial h_t}{\partial x}|_{x_t}$, updating in every step $t$. UKF does not pass the state distribution $x, P$ through the motion equations as a whole distribution, since such distribution transformation is unknown for general nonlinear state-transition. Instead, it samples *sigma points* from the original Gaussian distribution, passes each point through the nonlinear transition, and uses the resulting points to estimate the new Gaussian distribution. *Particle filters* go farther and do not bother to carry a Gaussian PDF: instead, the points themselves (*particles*) can be seen as representatives of the distribution.

Whether linear or not, a single simple model is hardly enough to represent the motion of any aerial target. A common approach is to maintain a switching model that repeatedly chooses between several mode, each controlled by a different motion model. Common simple models the Constant Velocity (CV), Constant Acceleration (CA), and Coordinated Turn Left or Right (CTL and CLR). Note that in all these models the prediction operator $F$ is linear. A popular switching mechanism is the *Interactive Multiple Model* (IMM) (Mazor et al., 1998), which relies on knowing the transition-probabilities between the various modes.

In addition to the challenge of predicting mode-transitions, the standard models in use are often too simple to represent the motions of modern and highly-maneuvering targets, such as certain unmanned aerial vehicles, drones and missiles. Many approaches have been recently attempted to improve tracking of such objects, as discussed in Section D.

### A.3 FORMULATION OF THE KALMAN FILTER MODEL FOR THE DOPPLER RADAR PROBLEM

Tracking targets from measurements of a Doppler radar is one of the main applications studied in this work. The basic setup of the problem is defined in Section 4.1. The KF model corresponding to this setup, in terms of Equation 1, is as follows:

$$F = \begin{pmatrix} 1 & & 1 & & \\ & 1 & & 1 & \\ & & 1 & & 1 \\ & & & 1 & \\ & & & & 1 \\ & & & & & 1 \end{pmatrix} \qquad H = H(X) = \begin{pmatrix} 1 & & & & & \\ & 1 & & & & \\ & & 1 & & & \\ & & & x_x/r & x_y/r & x_z/r \end{pmatrix} \qquad (4)$$

where $X = x_{full}$ (in the notation of Section 4.1) and $r = \sqrt{x_x^2 + x_y^2 + x_z^2}$. That is, the motion model is the standard constant-velocity model, and the observation model corresponds to direct observation of the location coordinates and a measurement of the radial component of velocity.

Both models ($F, H$) do not directly depend on the time. However, in contrast to the formulation of Equation 1, $H = H(X)$ does depend on the unknown state $X$, and thus $H(X)$ is unknown in inference time. That is, the observation model $h(X) = H(X) \cdot X$ is not linear anymore. The variants of the KF in Section 4 handle this non-linearity as follows:

1. **EKF** is inherently designed to handle such a non-linear model using the approximation $h(X) \approx \tilde{H} \cdot X$, where $\tilde{H} = \nabla_X h(\hat{X})$ and $\hat{X}$ is the recent estimate of $X$.

2. **KF**: for the standard KF, we use the common trick of approximating $H(X)$ with $\tilde{H} = H(z)$, where $z$ is the recent observation. That is, we replace the unknown location $(x_x, x_y, x_z)$ in Equation 4 with the location coordinates of the observation $(z_x, z_y, z_z)$, resulting in:

$$\tilde{H}(z) = \begin{pmatrix} 1 & & & & & \\ & 1 & & & & \\ & & 1 & & & \\ & & & z_x/\tilde{r} & z_y/\tilde{r} & z_z/\tilde{r} \end{pmatrix}$$

where $\tilde{r} = \sqrt{z_x^2 + z_y^2 + z_z^2}$. We also implemented other approximations (e.g., $\hat{X}$ instead of $z$), but they did not show added value for the KF baseline in our experiments.

The KF parameters $Q$ and $R$ were tuned by either Method 1 (noise-estimation) or Method 2 (filtering-errors optimization). Note that both methods are easily applicable to the non-linear setting:

1. Method 1: in Equation 2, we simply replace $H_t$ with the actual observation model $H(X)$ (note that $X$ *is* available in the training data in our settings).

2. Method 2: as before, we repeatedly run the KF and calculate the gradients of its loss. We simply use the KF (or EKF) implementation explained above.

### A.4   Recurrent Neural Networks

*Neural networks* (NN) are parametric functions, usually constructed as a sequence of matrix-multiplications with some non-linear differentiable transformation between them. NNs are known to be able to approximate complicated functions, given that the right parameters are chosen. Optimization of the parameters of NNs is a field of vast research for decades, and usually relies on gradient-based methods, that is, calculating the errors of the NN with relation to some training-data of inputs and their desired outputs, deriving the errors with respect to the network's parameters, and moving the parameters against the direction of the gradient.

*Recurrent neural networks* (RNN) (Rumelhart et al., 1986) are NN that are intended to be iteratively fed with sequential data samples, and that pass information (the *hidden state*) over iterations. Every iteration, the hidden state is fed to the next copy of the network as part of its input, along with the new data sample. *Long Short Term Memory* (LSTM) (Hochreiter and Schmidhuber, 1997) is an architecture of RNN that is particularly popular due to the linear flow of the hidden state over iterations, which allows to capture memory for relatively long term. The parameters of a RNN are usually optimized in a supervised manner with respect to a training dataset of input-output pairs.

### A.5   Noise Dependence by Change of Coordinates

One of the violations of the KF assumptions in Section 4 is the non-i.i.d nature of the noise. The interest in this violation is increased by the fact that the noise is simulated in an i.i.d manner, which makes the violation non-trivial to notice.

The violation is caused by the difference of coordinates: while the state of the target is represented in Cartesian coordinates, the noise is generated in spherical ones. As a result, spherical dimensions with higher noise are assigned to the same Cartesian dimensions in consecutive time-steps, creating dependence between the noise in these time-steps in Cartesian coordinates.

For example, consider a radar with a noiseless angular estimation (i.e., all the noise is radial), and consider a low target (in the same height of the radar). Clearly, most of the noise will be in the XY plane – both in the current time-steps and in the following ones (until the target gets far from the plane). In other words, the noise in the following time-steps statistically-depends on the current time-step, and is not i.i.d.

## B   Problem Setup and Implementation Details

**Code:**   The code for this work is available here, and naturally contains implementation details that may be missing in the descriptions below.

**Targets simulation:**   Our simulated dataset consists of episodes of tens to hundreds of seconds. Each episode contains around 20 targets. Each target appears at a certain point of time, has certain acceleration and speed ranges, and alternately performs straight motions (in constant speed or with acceleration) and turns (horizontally or vertically) until it disappears. The time progresses in discrete time-steps of size $dt$. The space is assumed to be 3D and homogeneous (no ground).

The randomization in an episode is expressed through the number of targets, each target's acceleration and speed ranges, target appearing time, initial target state, number of turns, directions of turns

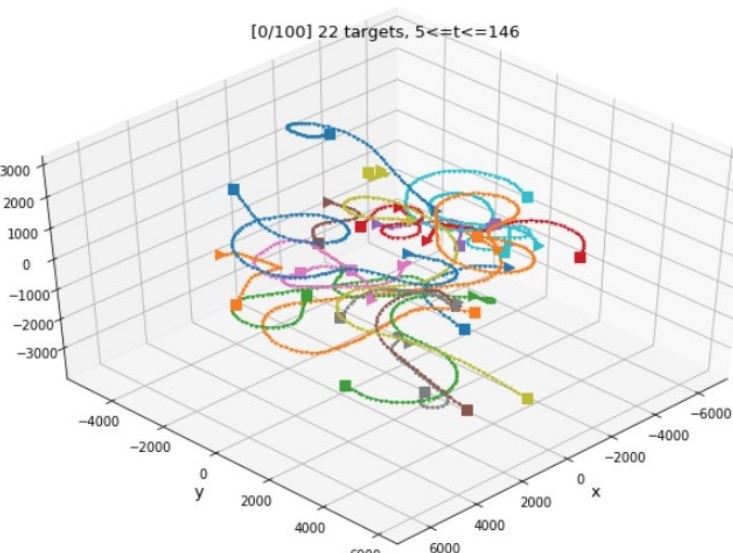

Figure 6: A sample of targets trajectories.

(left/right/up/down), length of turns, length of straight intervals, whether accelerating in straight intervals and what the acceleration is.

**Radar simulation:** Every time-step, the radar generates detections of the available targets. Each detection contains location information $(x, y, z)$ and a Doppler measurement corresponding to the projection of velocity onto the location vector. Each detection has an additive i.i.d Gaussian noise in spherical coordinates, i.e., range, azimuth, elevation, and Doppler. The radar itself is assumed to be a singular point located in the origin $(0, 0, 0)$. Recall that we assume a homogeneous space with no ground.

**Tracking test framework:** The predictive models (*trackers*) can be tested in a mere prediction framework (where each target is tested by itself), or in a full tracking framework (where multiple targets are tracked simultaneously). In the latter, we use a tracking system (*solver*) consisting of a matcher and multiple trackers. Every time-step during a test episode, the solver runs the following procedures:

1. **Observe**: receive observations from the radar.

2. **Assign**: use the matcher (based on the standard Hungarian Algorithm (Kuhn, 1955), with relation to the corresponding trackers' likelihood functions) to assign observations to trackers; create a new tracker for any unassigned observation; and delete any tracker whose target was not observed for several time-steps.

3. **Update**: update the location estimation of the trackers using the newly-assigned observations.

4. **Predict**: predict the target location in the next time-step (to allow more accurate assignment of the next batch of observations).

In the scope of this work we are mainly interested in the tracker module. Note that the tracker affects the matching accuracy through the likelihoods it feeds to the matcher; and affects the estimation accuracy directly in the prediction and update operations. To solve both matching and estimation tasks, the tracker is required to provide a good probabilistic model of the target location (namely, a PDF of the location) in addition to a point estimate.

We measure the estimation accuracy using the standard Mean Square Error (MSE) of the target location after the update step. For matching accuracy, for each target we consider the tracker that covers most of the target's observations, and count the percent of observations assigned to that tracker.

**Trackers:** To support the procedures described above, a tracker must implement prediction and update steps, as well as exporting the current point estimate of the target location and the likelihood of any observation whenever queried.

All the trackers in this work are conceptually based on Kalman Filter (described in Section A.2). The following elements characterize the exact nature of the tracker:

1. **KF/EKF**: the difference is in the observation model $H$, as explained in Section A.2.

2. **Coordinates for representation of the observation noise** $R$: while we represent the state in Cartesian coordinates $((x, y, z, vx, vy, vz)^\top)$ and measure the errors accordingly, the natural coordinates for the noise of the radar are spherical, as the observation errors are simulated independently for range, azimuth, elevation and Doppler. In particular, in Cartesian coordinates the errors are not independent, hence $R$ is not inherently constant. When implementing $R$ in spherical coordinates, we have to transform it to Cartesian coordinates every time-step according to the recent observation.

3. **Determining the noise parameters** $Q, R$ **by estimation/optimization**: by estimation of the noise we mean $Q = Cov(X_{t+1} - FX_t), R = Cov(Z_t - HX_t)$ over the training data $\{X_t, Z_t\}$. By optimization we mean that $R \in \mathbb{R}^{4\times 4}, Q \in \mathbb{R}^{6\times 6}$ are intended to minimize the loss of the predictions over the training data. Note that in the optimization, we represent $R, Q$ using lower-triangular decomposition so that both matrices remain SPD.

4. **Acceleration prediction**: replace the linear motion model $Fx$ with $Fx + 0.5a \cdot dt^2$, where $a$ is predicted using a LSTM that receives the observations and estimated states as recurrent inputs.

5. $Q$ **prediction**: dynamically predict the motion model noise $Q$ using an LSTM. We use the same LSTM as for $a$, except for the head layer that outputs $Q$ itself. For simplicity, we assume that $Q$ is diagonal when dynamically predicted.

6. $R$ **prediction**: dynamically predict the observation noise $R$ using a neural network that receives the recent observations and estimated states as inputs. For simplicity, we assume that $R$ is diagonal when dynamically predicted.

7. $H$ **prediction**: the observation model $z = h(x)$ of a radar is a non-linear function of $x$. Extended KF handles this through a linear approximation of $h$ around the currently-estimated state $x$. By $H$-prediction, we aim to learn to choose $H$ dynamically in an optimal way to improve the tracking errors.

8. **Mixture-of-Gaussians representation**: KF assumes Gaussian errors (in prediction and observation), hence the distribution of the state is always normal, which is somewhat limiting. We try to represent the distribution as a mixture of Gaussians (i.e., a linear combination of a few Gaussians). The Gaussians are intended to correspond to possible modes of motion (e.g., constant speed vs. a turn), and thus can predict the dynamic acceleration (see above) differently from each other. In addition, the amplitude of each Gaussian (i.e., the probability assigned to it) is predicted every step along with the accelerations.

Optimization of the noise parameters $(R, Q)$ was implemented as part of the Optimized Kalman Filter (OKF) described in Section 4. This feature was found to make the tracker both more accurate and more robust to design choices such as KF vs. EKF or coordinates of $R$ as described above. A neural-network-based Dynamic prediction of acceleration and noise was implemented as part of the Neural Kalman Filter (NKF) described in Section 5. The NKF architecture is also described in detail in Figure 7.

The last two features in the list failed to provide successful results: dynamic prediction of $H$ often yielded unstable predictions, and a mixture representation had insignificant effects on the prediction. Furthermore, beyond 2 Gaussians in the mixture, it seemed that any additional Gaussian became practically inactive, with amplitude being constantly near zero.

## B.1 TRAINING

**Training data:** A training dataset consists of targets, each consisting of a sequence of states (e.g., locations and velocities) along with a sequence of observations. In particular, the target state

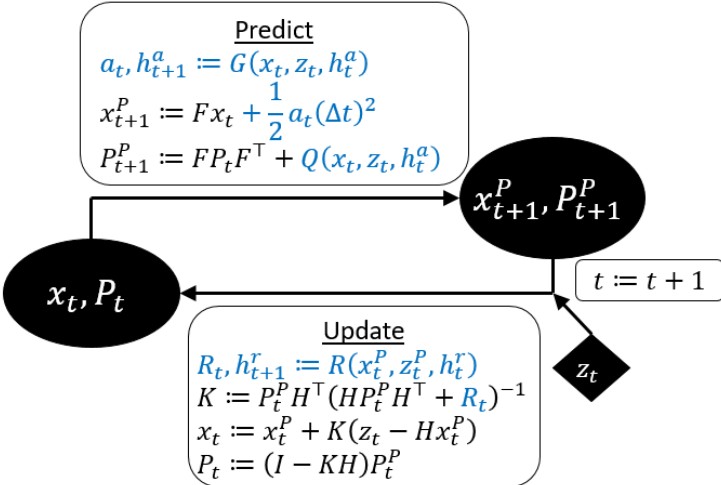

Figure 7: The Neural Kalman Filter (NKF) algorithm of Section 5. The differences from Figure 1 are marked. $G, Q, R$ are neural networks of the type LSTM, and $h^a, h^r$ are their hidden states (more accurately, $G$ and $Q$ are different heads of the same network, and thus share the same hidden states $h^a$). In addition to the raw $x_t, z_t$, the networks are also fed with certain manually-crafted features (e.g., the projection of $z_t - x_t$ on the perpendicular direction $x_t^\perp$).

ground-truth is assumed to be known and available to learn from. This can be achieved either by a simulated environment or by internal targets sensors (e.g., GPS) in a controlled experiment in a physical environment. Note that the training dataset considers the targets independently of each other, and does not refer to the notion of episodes with possibly-simultaneous targets. Targets assignment in such episodes can be handled by the Hungarian algorithm as mentioned above. The test dataset differs from the training dataset in the seeds used to generate the trajectories and the observations, and in certain experiments also in some configuration parameters such as the permitted range of acceleration.

**Noise estimation:** For models whose parameters are determined by noise estimation (e.g., the standard KF), the training reduces to calculation of $R$ and $Q$ from the data as the sample covariance matrices (Equation 2). Note that $\{x_t\}$ and $\{z_t\}$ from Equation 2 correspond to the states and observations mentioned above.

**Optimization (error minimization):** For models whose parameters are determined by optimization (e.g., OKF and NKF), we use the Adam algorithm (Diederik P. Kingma, 2014), as implemented by the PyTorch package, with initial learning rate of 0.01 (reduced by 50% every 150 training steps) and no weight-decay. Every training epoch goes once over each training target. In the beginning of each epoch, batches of 10 targets are randomly drawn.

Every training step corresponds to a single batch, where the training loss is aggregated additively over the targets in the batch and over the time-steps of each target. That is, for each target in the batch, we run the tracker on the target observations; after each observation receive the corresponding state estimate from the tracker; and add the corresponding error to the loss.

The explicit training loss is the Mean Square Error (MSE) of the estimated location (which is part of the estimated state) after the update step.

In the case study experiments on the Doppler radar domain (Section 4), each benchmark included 1500 simulated targets for training and 1000 targets for testing. A single training epoch was sufficient for convergence of the optimized models. The optimization lasted a few minutes per benchmark (scenario) – for all the 4 baselines in parallel (OKF, OKFp, OEKF, OEKFp).

## C  CHOLESKY GRADIENT OPTIMIZATION

Section 3 discusses an efficient method to apply gradient-based optimization to SPD matrices, based on parameterization of the SPD matrix using Cholesky decomposition (Pinheiro and Bates, 1996). This section presents this method more formally.

We consider an objective function $f(x; A)$ to be minimized, where $x$ is the input and $A \in \mathbb{R}^{n \times n}$ are parameters formed as a SPD matrix. We define $A(L) := LL^\top$ and

$$(L(\theta))_{ij} := \begin{cases} 0 & \text{if } i < j \\ e^{\theta_{n(n-1)/2+i}} & \text{if } i = j \\ \theta_{(i-2)(i-1)/2+j} & \text{if } i > j \end{cases}$$

where $\theta \in \mathbb{R}^{n(n+1)/2}$. We also denote the optimization update rule by $u(\theta, g)$, where $\theta$ are the current parameters and $g$ is the current gradient. For example, for the classic gradient descent, $u(\theta, g) = \theta - \alpha g$. Using these notations, the optimization method becomes as simple as Algorithm 1. This algorithm (with Adam's update rule) was used for all the KF optimizations in this work.

---

**Algorithm 1:** Cholesky Gradient Optimization

---

**Input**: objective $f$; update rule $u$; data $X$; initial parameters $\theta_0 \in \mathbb{R}^{n(n+1)/2}$;
**Output**: SPD matrix $A \in \mathbb{R}^{n \times n}$;
$\theta \leftarrow \theta_0$;
**for** training epoch **do**
    **for** data sample $x$ in $X$ **do**
        $g \leftarrow \nabla_\theta f(x; A(L(\theta)))$;
        $\theta \leftarrow u(\theta, g)$;
Return $A(L(\theta))$;

---

Note that the diagonal of $L(\theta)$ could be replaced by any increasing positive transformation. In fact, we could even drop the positive transformation and allow non-positive diagonal entries in $L$: this would only eliminate the uniqueness of the parameterization, and would allow $A$ to be positive *semi*-definite.

## D  RELATED WORK: EXTENDED

**Kalman Filter parameters tuning:**  Estimation of the noise parameters in KF has been researched for decades. A popular framework for this problem is estimation from data of observations $\{z\}$ alone (Odelson et al., 2006; Feng et al., 2014; Park et al., 2019), since the ground-truth of the states $\{x\}$ is often unavailable in the data (Formentin and Bittanti, 2014). In this work we assume that the ground-truth is available.

Many works addressed the problem of non-stationary noise estimation (Zanni et al., 2017; Akhlaghi et al., 2017). However, as demonstrated in Sections 4,5, in certain cases stationary methods are highly competitive if tuned correctly – even in problems with complicated dynamics.

Optimization of KF parameters with respect to tracking errors was already suggested in Abbeel et al. (2005), using a method that avoids gradients computation. In practice, "optimization" of KF parameters with respect to the errors is often handled manually using trial-and-error (Jamil et al., 2020), or using a grid-search over possible values of $Q$ and $R$ (Formentin and Bittanti, 2014; Coskun et al., 2017). Alternatively, the optimization is often simplified by restricting $Q$ and $R$ to be diagonal (e.g., see the experiments in Barratt and Boyd (2020) and Li et al. (2019)). Formentin and Bittanti (2014) pointed out explicitly that "since both the covariance matrices must be constrained to be positive semi-definite, $Q$ and $R$ are often parameterized as diagonal matrices".

Gradient-based optimization of SPD matrices in general was suggested in Tsuda et al. (2005) using matrix-exponents, and is also possible using projected gradient-descent (Tibshirani, 2015) – both rely on SVD-decomposition. In this work, we apply gradient-based optimization using the parameterization that was suggested in Pinheiro and Bates (1996), which requires a mere matrix multiplication, and thus is both efficient and easy to implement.

**Neural networks for tracking:** While comparison between RNN and KF goes back to DeCruyenaere and Hafez (1992); Chenna et al. (2004), the contribution of neural networks to non-linear motion tracking has been widely researched mostly in recent years, in the context of both radar and other sensors. Liu et al. (2019b) focus on an offline framework, where a bidirectional LSTM receives a full, normalized trajectory as an input, and improves the target locations estimations of a UKF in a 2-dimensional space. The LSTM is trained to predict the residual errors of the UKF. Kim et al. (2018) focus on a "near-online" framework, where assignment of an observation is decided after looking ahead a certain number of observations. They operate in the visual domain, incorporate both visual and motion information together to make decisions, and suggest a new variant of LSTM, where the hidden state $h$ interacts with the new input $x$ through the multiplication $hx$ rather than linear combination of the two, and thus can be interpreted as a dynamic model rather than hidden memory.

Gao et al. (2018) focus on an online framework, similarly to our work, where assignments and estimations are done immediately with the receipt of the observations. For the private case of a one-dimensional recurrent input signal, they use two LSTMs for estimation of the target location – one for the prediction step and one for the update step – to achieve improved estimation. Gao et al. (2019) expand this work and learn to predict both the location estimate and its (one-dimensional) variance according to the negative-log-likelihood loss, which yields similar accuracy but provides uncertainty estimation. Dan Iter (2016) consider online tracking of single vehicles on the 2-dimensional video screen of a vehicle camera (the KITTI object tracking benchmark). Since they assume zero observation error, they focus on prediction error optimization. Their LSTM yields a better point-prediction than KF, but does not improve the end-to-end tracking, which they hypothesize that could be caused by insufficient uncertainty estimation. Coskun et al. (2017) suggest a combination of LSTM and KF for the task of human pose estimation, learned on basis of the square estimation-errors. In particular, three LSTM modules are used to predict the parameters $F, Q, R$ of a KF.

Liu et al. (2019a) train an LSTM for data association, i.e., the assignment task. Sengupta et al. (2019) address the problem of sensors fusion using a deep neural network. They incorporate information from a monocular camera and a mmWave radar, and show how to handle failures in either of the sensors. Deng et al. (2020) consider standard models as building blocks for IMM, and replace the mode-transition matrix with an LSTM that estimates the transition probabilities dynamically. Peng and Gu (2011) replace the standard building blocks of IMM with modes that are dedicatedly-designed to track highly-maneuvering targets.

A combination of KF with attention models (Vaswani et al., 2017) was also suggested (Liu et al., 2020) in the context of click-through rate prediction in e-commerce, where vital information for prediction may be concentrated in certain time-steps on which the model should focus.

# E  THEORETICAL ANALYSIS: ESTIMATION VS. OPTIMIZATION IN THE TOY BENCHMARK

Section 4 demonstrates that the optimal noise parameters in KF often differ from the covariance of the noise. Here we analyze this effect for the private case of the Toy benchmark, where all KF assumptions (Definition 2.1) hold – except for the non-linear observation model $H(X)$.

**Definition E.1** (The toy Doppler problem). The toy Doppler problem is the filtering problem modeled by Equation 1 with the system-state space $X = (x, y, z, u_x, u_y, u_z)^\top \in \mathbb{R}^6$ and the following parameters:

$$F = \begin{pmatrix} 1 & & & 1 & & \\ & 1 & & & 1 & \\ & & 1 & & & 1 \\ & & & 1 & & \\ & & & & 1 & \\ & & & & & 1 \end{pmatrix} \qquad H = H(X) = \begin{pmatrix} 1 & & & & & \\ & 1 & & & & \\ & & 1 & & & \\ & & & x/r & y/r & z/r \end{pmatrix}$$

$$Q = \mathbf{0} \in \mathbb{R}^{6 \times 6} \qquad R = \begin{pmatrix} \sigma_x^2 & & & \\ & \sigma_y^2 & & \\ & & \sigma_z^2 & \\ & & & \sigma_D^2 \end{pmatrix}$$

where $r = \sqrt{x^2 + y^2 + z^2}$ and $\sigma_x, \sigma_y, \sigma_z, \sigma_D > 0$.

As mentioned in Section 4, this toy problem is highly simplified. For example, the targets only move in exact straight lines, and the noise of the radar is drawn i.i.d in *Cartesian* coordinates. Yet, despite the major simplifications, we show below that the remaining violation of observation linearity leads to sub-optimality of noise estimation when determining the KF parameters.

Since the true $X$ is unknown to the KF in the inference phase, we assume that the KF approximates $H(X)$ using $\tilde{H}$, which relies on the current estimates of $x, y, z$:

$$\tilde{H} = \begin{pmatrix} 1 & & & \\ & 1 & & \\ & & 1 & \\ & & & \hat{x}/\hat{r} & \hat{y}/\hat{r} & \hat{z}/\hat{r} \end{pmatrix}$$

**Lemma E.1** (Effective observation noise in the toy Doppler problem). Consider the toy Doppler problem with the approximated observation model $\tilde{H}$. At a certain time-step, assume that the positional estimation errors (i.e., the estimation errors of $x, y, z$) are independent of the target velocity $u$. Then, the observation noise corresponding to $\tilde{H}$ (the *effective* observation noise) is described by the following covariance matrix:

$$\tilde{R} = \begin{pmatrix} \sigma_x^2 & & & \\ & \sigma_y^2 & & \\ & & \sigma_z^2 & \\ & & & \sigma_D^2 + C \end{pmatrix} = R + \begin{pmatrix} 0 & & & \\ & 0 & & \\ & & 0 & \\ & & & C \end{pmatrix} \tag{5}$$

for some $C > 0$.

*Proof.* Denoting $\tilde{x} := x/r$ (and similarly for $\tilde{y}, \tilde{z}$), we can rewrite $\tilde{H}$ as

$$\tilde{H} = \begin{pmatrix} 1 & & & \\ & 1 & & \\ & & 1 & \\ & & & \tilde{x} + d\tilde{x} & \tilde{y} + d\tilde{y} & \tilde{z} + d\tilde{z} \end{pmatrix}$$

where $d\tilde{x}, d\tilde{y}, d\tilde{z}$ are the corresponding (normalized) estimation errors. Note that in the private case where $x, y, z$ are estimated by the observations $z_x, z_y, z_z$, these errors coincide with the corresponding entries of the observation noise $\nu$ (up to the normalization by $r$).

By shifting the observation model in Equation 1 from $H$ to $\tilde{H}$, we receive

$$Z = H \cdot X + \nu = \tilde{H}X + \begin{pmatrix} \nu_x \\ \nu_y \\ \nu_z \\ \nu_D - d\tilde{x}u_x - d\tilde{y}u_y - d\tilde{z}u_z \end{pmatrix} = \tilde{H}X + \begin{pmatrix} \nu_x \\ \nu_y \\ \nu_z \\ \nu_D - d\tilde{r}^\top \cdot u \end{pmatrix}$$

that is, the observation noise corresponding to $\tilde{H}$ is $\tilde{\nu} = (\nu_x, \nu_y, \nu_z, \nu_D - d\tilde{r}^\top \cdot u)^\top$.

$d\tilde{x}, \nu_x$ are both independent of the velocity $u_x$ (the former by the lemma's assumption, and the latter by the model of Equation 1 in Definition E.1). Thus, $Corr(d\tilde{x} \cdot u_x, \nu_x) = E(d\tilde{x} \cdot u_x \cdot \nu_x) = E(d\tilde{x}\nu_x)E(u_x) = 0$, and similarly for $\nu_y, \nu_z$. Hence, by denoting $C = Var(d\tilde{r}^\top \cdot u) > 0$, the covariance matrix of $\tilde{\nu}$ is indeed $\tilde{R}$. $\qquad\square$

Lemma E.1 provides the theoretical explanation for Figure 4b in Section 4, where we see that the optimization of $R$ increases the variance of the Doppler signal compared to the variance of the positional signal. Figure 4b also shows a decrease in the positional variance (not only an increase in Doppler's variance), which is not explained by the analysis above. This is caused by the fact that when $Q \equiv 0$, the absolute values of $R$ have quite minor importance compared to the relative values between the components – hence the optimization increases Doppler's variance compared to positional variance, but is quite indifferent to their scale. Indeed, re-scaling of the optimized $R$ provided indistinguishable empirical results.

Note that in this private case of Toy benchmark, the gap between estimation and optimization could be easily solved by modifying the estimation from $R := Cov(\{z_t - H(x_t) \cdot x_t\})$ to $R := Cov(\{z_t - H(z_t) \cdot x_t\})$. However, this correction does not guarantee optimal results beyond the Toy benchmark, and in particular did not demonstrate significant improvement in the other benchmarks defined in Section 4. Indeed, while certain private cases can be modeled and solved analytically through effective noise estimation, optimization of the parameters provides a more robust solution.

# F    THEORETICAL ANALYSIS: ESTIMATION VS. OPTIMIZATION IN THE PRESENCE OF NON-I.I.D NOISE

The KF model assumes i.i.d noise for both the motion model and the observation model. In cases where the noise is known to follow another specific model, a dedicated solution can sometimes be developed. For example, if the noise is auto-regressive with a known order $p$, an adjusted KF model may consider the last $p$ values of the noise itself as part of the system state (Geist and Pietquin, 2011). However, we often do not know the actual model of the noise, or we do not know how to develop the corresponding analytical solution, or we are not aware at all of the violation of the i.i.d assumption. In any of these cases, the default practical choice is to keep the original KF model. However, the covariance matrices of the noise are not guaranteed anymore to be the optimal values of the KF parameters $Q$ and $R$.

For example, consider the problem of a moving target in a two-dimensional space, with observation noise drawn i.i.d in polar coordinates (similarly to the lidar model of Appendix J). For simplicity of the analysis we represent the state using only the location $x = (x_1, x_2)^\top \in \mathbb{R}^2$, and assume a model of no-motion up to an isotropic noise. Furthermore, we assume that the observation noise has only a radial component.

**Definition F.1** (The simplified lidar problem). The simplified lidar problem is the filtering problem modeled by Equation 1 with the following parameters:

$$F = H = \begin{pmatrix} 1 & 0 \\ 0 & 1 \end{pmatrix} \qquad Q = \begin{pmatrix} q & 0 \\ 0 & q \end{pmatrix} \qquad R_{polar} = \begin{pmatrix} r_0 & 0 \\ 0 & 0 \end{pmatrix}$$

for some $q, r_0 > 0$, where the observation noise $\nu_t$ is drawn i.i.d from $N(0, R_{polar})$ in *polar* coordinates. We also assume that the initial system-state $x_0$ is drawn from an isotropic distribution, i.e., follows a density function of the form $f((x_0)_1^2 + (x_0)_2^2)$, which is invariant to the target direction.

At a first glance, this simplistic problem may seem to satisfy the KF assumptions. In particular, both sources of noise are i.i.d. However, the observation noise is drawn i.i.d in *polar* coordinates, while the system state is estimated (and the prediction error is measured) in *Cartesian* coordinates. As discussed in Appendix A.5, the i.i.d property is *not* invariant to the coordinates transformation!

**Lemma F.1.** The observation noise in the simplified lidar problem is not i.i.d in Cartesian coordinates.

*Proof.* Denote the system state at time $t$ by $x_t = ((x_t)_1, (x_t)_2)^\top$, and denote $\tan \theta = ((x_t)_2/(x_t)_1)$. From Definition F.1, by direct coordinates transformation, the observation noise is drawn from the distribution $\nu_t \sim N(0, R(\theta))$, where

$$R(\theta) = \begin{pmatrix} r_0 \cos^2(\theta) & r_0 \cos(\theta) \sin(\theta) \\ r_0 \cos(\theta) \sin(\theta) & r_0 \sin^2(\theta) \end{pmatrix}$$

This noise covariance matrix varies every time-step with $\theta$, hence the noise $\nu_t$ is not identically-distributed in Cartesian coordinates. Furthermore, consecutive time-steps are likely to have similar values of $\theta$ (according to the motion model), hence the noise is not independent either.    □

If we are aware of the particular noise model of the problem, we can estimate $R_{polar}$ in polar coordinates and transform it to the Cartesian $R(\theta)$ dynamically every time-step (e.g., using our the current estimate of $\theta$, since the true $\theta$ is unknown). This is indeed what we do in the KFp model in Section 4. If we are not aware, however, we would simply estimate a constant covariance matrix $R$ in Cartesian coordinates.

**Lemma F.2** (Noise estimation in the simplified lidar problem). Let $R$ be the sample covariance matrix of the observation noise, calculated by Equation 2 with respect to a dataset of i.i.d targets in the simplified lidar problem. Then, as the number of targets in the data grows, $R$ converges almost surely to

$$R_{est} = \begin{pmatrix} r_0/2 & 0 \\ 0 & r_0/2 \end{pmatrix}$$

*Proof.* From Definition F.1, the distribution of the initial system-states is isotropic, and so is the motion model $Q$. Hence, for any target at any time-step, the distribution of its direction $\theta$ (where $\tan\theta = ((x_t)_2/(x_t)_1)$) follows the uniform distribution: $\theta \sim [0, 2\pi]$. Now we can calculate:

$$E_\theta\left[(R_{est})_{11}\right] = E_\theta\left[r_0 \cos^2\theta\right] = \int_0^{2\pi} \frac{r_0}{2\pi}\cos^2\theta d\theta = \frac{r_0}{2}$$

$$E_\theta\left[(R_{est})_{22}\right] = E_\theta\left[r_0 \sin^2\theta\right] = \int_0^{2\pi} \frac{r_0}{2\pi}\sin^2\theta d\theta = \frac{r_0}{2}$$

$$E_\theta\left[(R_{est})_{12}\right] = E_\theta\left[(R_{est})_{21}\right] = E_\theta\left[r_0 \cos\theta\sin\theta\right] = \int_0^{2\pi} \frac{r_0}{2\pi}\cos\theta\sin\theta d\theta = 0$$

and since the targets are i.i.d, by the Law of Large Numbers we have

$$R \xrightarrow{\text{a.s.}} R_{est} = \begin{pmatrix} r_0/2 & 0 \\ 0 & r_0/2 \end{pmatrix}$$

$\square$

So far we saw that the actual observation noise is $R(\theta)$ (where $\theta$ changes every time-step), while the constant covariance matrix that would be obtained by noise estimation is (asymptotically) $R_{est}$. Assuming that we constraint ourselves to a constant matrix (invariant to $\theta$) in Cartesian coordinates, can we do better than $R_{est}$? More specifically, due to the radial symmetry of the problem, we can denote w.l.o.g

$$R_{opt}(r) = \begin{pmatrix} r & 0 \\ 0 & r \end{pmatrix}$$

for some $r > 0$, and ask what value of $r$ would result in minimal square filtering errors of the KF.

**Lemma F.3** (Optimization in the simplified lidar problem). Assume that at a certain time-step in the simplified lidar problem, the state of the system is known to be normally-distributed with mean and covariance

$$x_0 = (x_1, x_2)^\top \qquad P_0 = \begin{pmatrix} p & 0 \\ 0 & p \end{pmatrix}$$

and that a new observation $z = (x_1 + dx_1, x_2 + dx_2)^\top$ is received. Denote by $x$ the point-estimate of the KF following a filtering step with respect to noise parameters of the form $R_{opt}(r)$. Then, $r = \frac{pr_0}{2p+r_0}$ minimizes the expected square error ($MSE$) of $x$. Furthermore, $(R_{opt})_{ii} = r < (R_{est})_{ii}$ (for $i \in \{1, 2\}$), i.e., the optimal value of $r$ is smaller than the value obtained by noise estimation.

*Proof.* As the true model of the observation noise is $R(\theta)$, by applying the KF update step (specified in Figure 1) we obtain the new true mean of the system-state distribution:

$$x_{true} = x_0 + P_0 H^\top (H P_0 H^\top + R)^{-1}(z - H x_0) = x_0 + P_0(P_0 + R)^{-1}(z - x_0)$$

$$= x_0 + \begin{pmatrix} \frac{r_0 \sin^2\theta + p}{p+r_0} & -\frac{r_0 \cos\theta\sin\theta}{p+r_0} \\ -\frac{r_0 \cos\theta\sin\theta}{p+r_0} & \frac{r_0 \cos^2\theta + p}{p+r_0} \end{pmatrix} \begin{pmatrix} dx_1 \\ dx_2 \end{pmatrix} = \begin{pmatrix} x_1 + \frac{(r_0 \sin^2\theta + p)dx_1 - (r_0 \cos\theta\sin\theta)dx_2}{p+r_0} \\ x_2 + \frac{(r_0 \cos^2\theta + p)dx_2 - (r_0 \cos\theta\sin\theta)dx_1}{p+r_0} \end{pmatrix}$$

where $\theta$ corresponds to the (unknown) true direction of the target. However, due to the restricted form of $R_{opt}(r)$ (which is independent of $\theta$), the actual estimate would be

$$x = (x_1 + \frac{p}{p+r}dx_1, x_2 + \frac{p}{p+r}dx_2)^\top$$

The expected error of the point-estimate $x$ can be decomposed into two terms: the bias of $x$, and the variance of the system-state distribution. More formally, we can write $MSE = MSE_{var}(P_{true}) + MSE_{bias}(x, x_{true})$, where we can only control the latter. Specifically, we denote $a(r) := p/(p+r)$

and look for the value of $a$ that minimizes the $MSE$. We use the identities $\sin 2\theta = 2\cos\theta\sin\theta$ and $E\left[dx_1^2\right] = E\left[dx_2^2\right] = q$ (the latter is by the motion model in Definition F.1).

$$MSE_{bias}(x(a), x_{true}) = E||x - x_{true}||^2$$

$$= E\left[\left((a - \frac{r_0\sin^2\theta + p}{p + r_0})dx_1 + \frac{r_0\sin(2\theta)/2}{p + r_0}dx_2\right)^2\right.$$

$$+ \left.\left((a - \frac{r_0\cos^2\theta + p}{p + r_0})dx_2 + \frac{r_0\sin(2\theta)/2}{p + r_0}dx_1\right)^2\right]$$

$$= E\left[dx_1^2\left(a^2 - 2a\frac{r_0\sin^2\theta + p}{p + r_0} + C_1\right) + \frac{r_0^2\sin^2(2\theta)/4}{(p + r_0)^2}dx_2^2 + A_1dx_1dx_2\right.$$

$$+ \left.dx_2^2\left(a^2 - 2a\frac{r_0\cos^2\theta + p}{p + r_0} + C_2\right) + \frac{r_0^2\sin^2(2\theta)/4}{(p + r_0)^2}dx_1^2 + A_2dx_1dx_2\right]$$

$$= 2qa^2 - 2qa\frac{r_0 + 2p}{p + r_0} + q(C_1 + C_2) + q\frac{r_0^2\sin^2(2\theta)/2}{(p + r_0)^2}$$

where $C_{1,2}$ are independent of $a$, and $A_{1,2}$ are multiplied by $E[dx_1dx_2] = 0$ and vanish. To minimize we calculate

$$0 = \frac{\partial MSE_{bias}(x(a), x_{true})}{\partial a} = 4q \cdot a - 2q\frac{2p + r_0}{p + r_0}$$

which gives us

$$a = \frac{p + r_0/2}{p + r_0}$$

Note that the expression of $MSE_{bias}$ above clearly diverges as $|a| \to \infty$, hence the only critical point necessarily corresponds to a minimum of the $MSE$. Thus, the optimal $MSE$ is given when substituting the following $r$ in $R_{opt}$:

$$r = p/a - p = \frac{p^2 + pr_0 - (p^2 + pr_0/2)}{p + r_0/2} = \frac{pr_0}{2p + r_0}$$

For the last part of the lemma, we recall that $(R_{est})_{ii} = r_0/2$ and compare to $r$ directly:

$$(R_{est})_{ii} - (R_{opt})_{ii} = r_0/2 - r = \frac{(pr_0 + r_0^2/2) - pr_0}{2p + r_0} = \frac{r_0^2/2}{2p + r_0} > 0$$

$\square$

Lemma F.3 shows that independently of the value of $x_0$ (i.e., independently of the current state of the system or the current direction of the target), the expected filtering error would benefit from reduction of the noise parameter to $(R_{opt})_{ii} = r < (R_{est})_{ii}$. Hence, noise estimation is not an optimal strategy for parameters tuning in this problem.

The simplified lidar problem can be seen as a simplification of the problem experimented in Appendix J, with a degenerated motion model. In the experiments, the optimization of the KF reduced the $MSE$ by 14.9%, and the learned parameters indeed decreased the observation noise, as shown in Figure 8.

In summary, in this section we introduce a simplified lidar problem. While at first glance the problem seems to follow the KF assumptions, the i.i.d-noise assumption is in fact violated by the natural polar coordinates of the sensor. Consequentially, the optimal values of the $R$ parameters do not correspond anymore to the covariance of the observation noise, but rather should be reduced. In other words, **the non-i.i.d nature of the noise reduces the *effective* noise**. This theoretic result is consistent with the experiments presented in Appendix J.

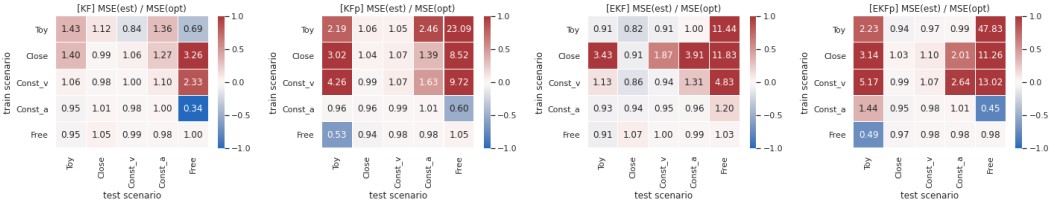

Figure 8: The observation noise ($R$) and prediction noise ($Q$) matrices obtained by noise estimation (KF) and by optimization (OKF), for the problem of state-estimation from lidar measurements (described in Appendix J).

## G  GENERALIZATION TESTS: TRAINING AND TESTING ON DIFFERENT BENCHMARKS

In the case-study of Section 4, we demonstrate the robustness of an Optimized KF (OKF) over a variety of benchmarks representing different tracking scenarios – where in every benchmark, OKF (Method 2) obtained lower estimation errors than a standard KF (Method 1), **over an out-of-sample test data**. This demonstrates that OKF did not overfit the training data, but still relies on representativeness of the training data, i.e., on the assumption that the training data and the test data are taken from the same distribution.

A stronger generalization ability is demonstrated in Section 5, where OKF beats both KF and NKF not only on out-of-sample test data with the same settings, but also **on a test data with different settings**. Specifically, we consider changes of factors 0.5 and 2 in the scale of the targets acceleration. This essentially changes the motion of the targets, including turns sharpness, speed changes and typical scale of speed. In terms of the linear motion model of KF, the acceleration corresponds to the motion noise $Q$, hence we essentially changed the noise after it had been learned, which poses a significant generalization challenge. Thus, the results of Section 5 provide a significant evidence for the robustness of OKF to certain distributional shifts.

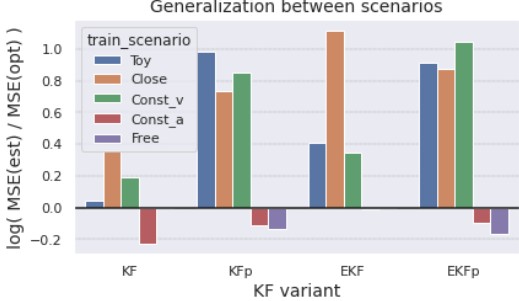

(a) $MSE\_ratio = MSE(KF)/MSE(OKF)$ for every KF-baseline (KF,KFp,EKF,EKFp defiend in Section 4), and for every pair of train-scenario and test-scenario. The colormap is logarithmic ($\sim log(MSE\_ratio)$), where red values represent advantage to OKF ($MSE\_ratio > 1$).

(b) For every train-scenario, $MSE\_ratio$ is averaged over all the test-scenarios and is shown in a logarithmic scale. Positive values indicate advantage to OKF.

Figure 9: Generalization tests: optimization vs. noise estimation under distributional shifts.

In this section we present a yet stronger evidence for the robustness of OKF – not over a parametric distributional shift, but **over entirely different benchmarks**. Specifically, we consider the

5 benchmarks (or scenarios) of Section 4. For every pair (train-scenario, test-scenario), we train both KF and OKF on data of the train-scenario, then test them on data of the test-scenario. For every such pair of scenarios, we measure the generalization advantage of OKF over KF through $MSE\_ratio = MSE(KF)/MSE(OKF)$ (where $MSE\_ratio > 1$ indicates positive advantage). To measure the total generalization advantage of a model trained on a certain scenario, we calculate the geometric mean of $MSE\_ratio$ over all the test-scenarios (or equivalently, the standard mean over the logs of the ratios). Note that the logarithmic scale is indeed more natural for a symmetric view of the ratio of the two MSE scores.

This test is quite noisy, since a model optimized for a certain scenario may legitimately be inferior in other scenarios. Yet, considering all the results together in Figure 9, it is evident that optimization of KF yields more robust models in most cases, and even when it *is* inferior to noise-estimation, it does not lose by a large margin.

## H    Sensitivity of KF-Optimization to Train Dataset Size

Each benchmark in the case-study of Section 4 has 1500 targets in its train data. Training from these datasets demonstrated significant advantage to optimization of KF over noise-estimation. However, one may argue that on smaller datasets, optimization may struggle harder than noise-estimation, e.g., due to numeric instability with little data. Furthermore, the standard optimization procedure "wastes" a portion of the train data as a validation set, which does not contribute directly to the training, and thus may increase the sensitivity to the amount of data.

To test the sensitivity of KF optimization to the amount of data, we repeated some of the tests of Section 4 using smaller subsets of the train datasets – beginning from as few as 20 targets per dataset. Figure 10 demonstrates that the advantage of OKF over KF holds consistently over the various sizes of the train datasets, although it indeed increases with the amount of data. In fact, in the Free Motion benchmark, the state-estimation errors of the standard KF monotonously *increase* with the amount of data.

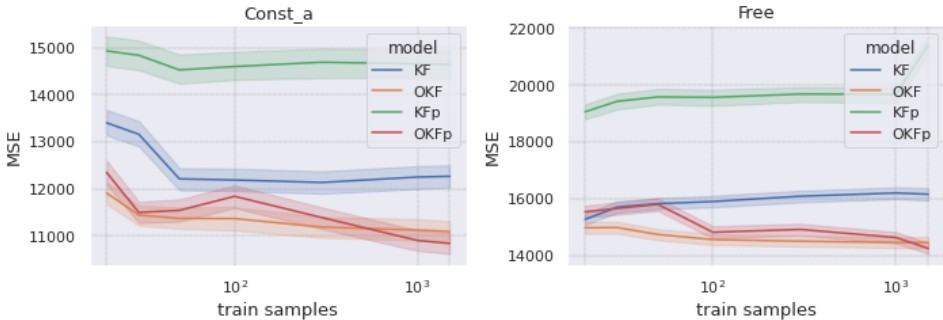

Figure 10: The advantage of OKF over KF remains positive – although sometimes smaller – even when the train data reduces to as few as 20 targets. The shadowed areas correspond to 95% confidence-intervals.

## I    Optimized Kalman Filter for Video Tracking

Radar tracking is an interesting domain with unique properties such as non-linear observation model and radial observation noise. In this domain, Section 4 demonstrates the sub-optimality of KF with estimated noise parameters, compared to a KF with optimized parameters. In this section, we compare the two in the popular and arguably-simpler domain of video tracking, using the MOT20 dataset (Dendorfer et al., 2020) (available under *Creative Commons Attribution-NonCommercial-ShareAlike 3.0* License).

MOT20 includes several videos with multiple targets (mostly pedestrians) to track. The dataset also includes the ground-truth location and size of the targets in every frame of the videos. We consider these ground-truth states as direct observations, i.e., we assume to have a detector with zero observation error (which in particular makes the observation model linear, in contrast to the radar tracking problem).

More specifically, we characterize a target using the state $X = (x, y, w, h, vx, vy) \in \mathbb{R}^6$ (two-dimensional location, size and velocity), and an observation as $Z = (x, y, w, h)$. Note that $x, y, w, h$ are provided in the dataset and we derive $vx, vy$ from $x, y$. The observation model $H$ is known as described above, and for the motion model $F$ we assume constant velocity and constant target size, leading to:

$$
H = \begin{pmatrix} 1,0,0,0,0,0 \\ 0,1,0,0,0,0 \\ 0,0,1,0,0,0 \\ 0,0,0,1,0,0 \end{pmatrix}, F = \begin{pmatrix} 1,0,0,0,1,0 \\ 0,1,0,0,0,1 \\ 0,0,1,0,0,0 \\ 0,0,0,1,0,0 \\ 0,0,0,0,1,0 \\ 0,0,0,0,0,1 \end{pmatrix}
$$

$Q$ and $R$ are estimated or optimized with relation to the train data, as described in Section 4. For the train data we use MOT20-01,MOT20-02,MOT20-03, and for the test data MOT20-05. In particular, our test data comes from an entirely different video than the train data, hence the testing is less prone to overfit. The goal function for the optimization is the MSE (mean square error) of the location $x, y$ after the prediction step (note that in absence of observation noise, the error of the update step is simply zero).

As shown in Figure 3, the optimized model (OKF) reduces the prediction errors (MSE) by 18% in comparison to a KF with standard noise estimation. The significance level of the results is also indicated by the huge z-value $z = 125$, corresponding to the errors difference (calculated as $z = mean(\Delta err)/std(\Delta err) \cdot \sqrt{N_{samples}}$). The results indicate that the sub-optimality of noise estimation – as well as the benefits of parameters optimization – are a cross-domain phenomenon and are not limited to radar tracking or to non-linear observation models.

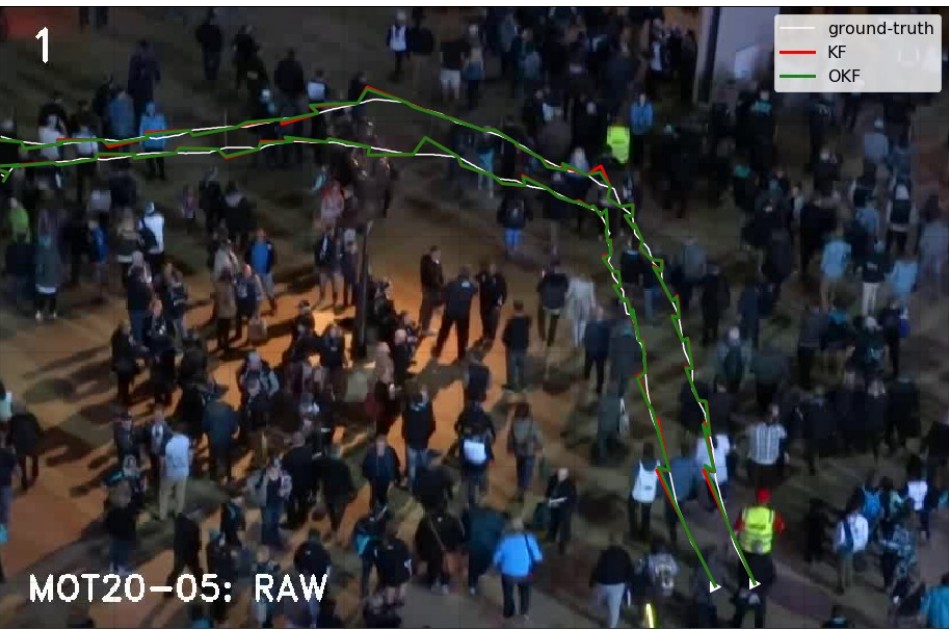

Figure 11: A sample of 2 targets in the first frame of the test video in MOT20. The true trajectories of the targets are shown along with the predictions of KF and OKF (each prediction is done one time-step in advance).

## J  OPTIMIZED KALMAN FILTER FOR LIDAR-BASED STATE ESTIMATION IN SELF DRIVING

Another domain in which we test the Optimized Kalman Filter (OKF) of Section 4 is state-estimation in self-driving based on lidar measurements with respect to known landmarks (Moreira et al., 2020). For simplicity, we focus on location estimation and on a single landmark (as the matching problem from multiple landmarks is out of the scope of this work).

We formulate the problem as follows: the target state is $X = (x, y, vx, vy) \in \mathbb{R}^4$ (two-dimensional location and velocity), and an observation is $Z = (x, y)$. The dynamics are simulated as follows: each target trajectory consists of several intervals, in each one the target has certain (positive or negative) acceleration and certain lateral acceleration ("turn magnitude"), both drawn randomly at the beginning of the interval. The observation noise is drawn i.i.d in polar coordinates. We simulated a corresponding dataset of 2000 targets as demonstrated in Figure 12a, and split them to train set (1400 targets) and test set (600 targets).

From the KF point-of-view, the dynamics and observations are modeled by:

$$H = \begin{pmatrix} 1, 0, 0, 0 \\ 0, 1, 0, 0 \end{pmatrix}, F = \begin{pmatrix} 1, 0, 1, 0 \\ 0, 1, 0, 1 \\ 0, 0, 1, 0 \\ 0, 0, 0, 1 \end{pmatrix}$$

Note that the lidar problem is inherently different from the radar problem (Section 4) since the observation model is linear (due to the absence of the Doppler signal); and is different from video tracking (as formulated in Appendix I) since the observations are not assumed to be noiseless. On the other hand, similarly to the case of the radar, the Cartesian representation of the observations prevents their noise from being i.i.d, since the observation noise is only i.i.d in polar coordinates (see Appendix A.5).

We trained the OKF of Section 4 with respect to the MSE of the estimated target location, as described in Appendix B.1; and trained a standard KF by estimating the noise as in Equation 2. Then we tested both models on the test data. The optimization significantly reduced the errors by 14.9%, as shown in Figure 12b. This joins the promising results from the two other domains (radar tracking and video tracking), and demonstrates the generality of our method.

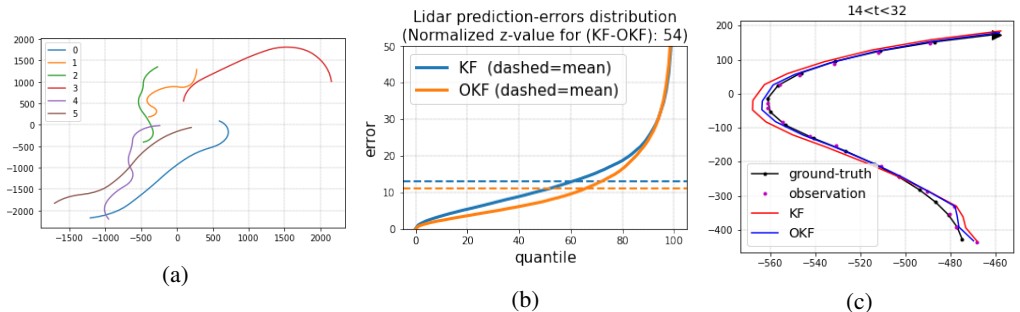

(a)  (b)  (c)

Figure 12: (a) A sample of 6 targets trajectories from the simulated self-driving data. (b) Prediction errors of KF and OKF on 600 targets of the test data. OKF is clearly more accurate, as also indicated by the huge z-value corresponding to the errors difference (calculated as $z = mean(\Delta err)/std(\Delta err) \cdot \sqrt{N}$). (c) the tracking of KF and OKF on a sample target trajectory, focused on the time interval $14 < t < 32$ for visual clarity.

## K  NEURAL KALMAN FILTER: EXTENDED EXPERIMENTS

Section 5 demonstrates that even if a neural-network (NN) based model provides more accurate predictions than a standard KF – its advantage may be entirely vanished when compared to an optimized KF instead. It is shown for the specific NKF model in a free-motion benchmark in the Doppler radar problem. This is sufficient to demonstrate the error in the common methodology that compares optimized NN-based models to non-optimized variants of KF.

To show that these results are not exclusive to the experiments of Section 5, we present extended experiments in this section. Specifically, we consider 2 different benchmarks – the free-motion benchmark of Section 5 and Const_a benchmark of Section 4 (in which the targets may have acceleration but not turns). We also consider 3 different neural models:

- Predicted-acceleration KF (aKF): a variant of NKF without dynamic prediction of the covariance matrices $Q$ and $R$ in every step (in terms of Appendix B: a tracker with elements 3, 4).

- Neural KF (NKF): the model used in Section 5 and shown in Figure 7 (in terms of Appendix B: a tracker with elements 4, 5, 6).

- Neural KF with H-prediction (NKFH): a variant of NKF that also predicts the observation model $H$ dynamically in every step (in terms of Appendix B: a tracker with elements 4, 5, 6, 7).

For each benchmark and each model, we trained the model on train data with a certain range of targets acceleration (note that acceleration affects both speed changes and turns sharpness), and tested it on targets with three different acceleration ranges – the original one, a smaller one and a larger one. That is, some of the test datasets correspond to distributional shifts. For each model we train two variants – one with Cartesian representation of the observation noise $R$, and one with spherical representation (as in the baselines of Section 4) – and we select the best among the two variants using the MSE scores on the validation data (which is a portion of the data assigned for training).

We measure the results using (1) the Mean Square Error (MSE) of the estimated location after each update step; and (2) the Negative-Log-Likelihood (NLL) of the true system-state with relation to the estimated state-distribution, after the prediction step. Note that NLL is important for matching observations to targets in the multi-target tracking problem (which is out of the scope of this work).

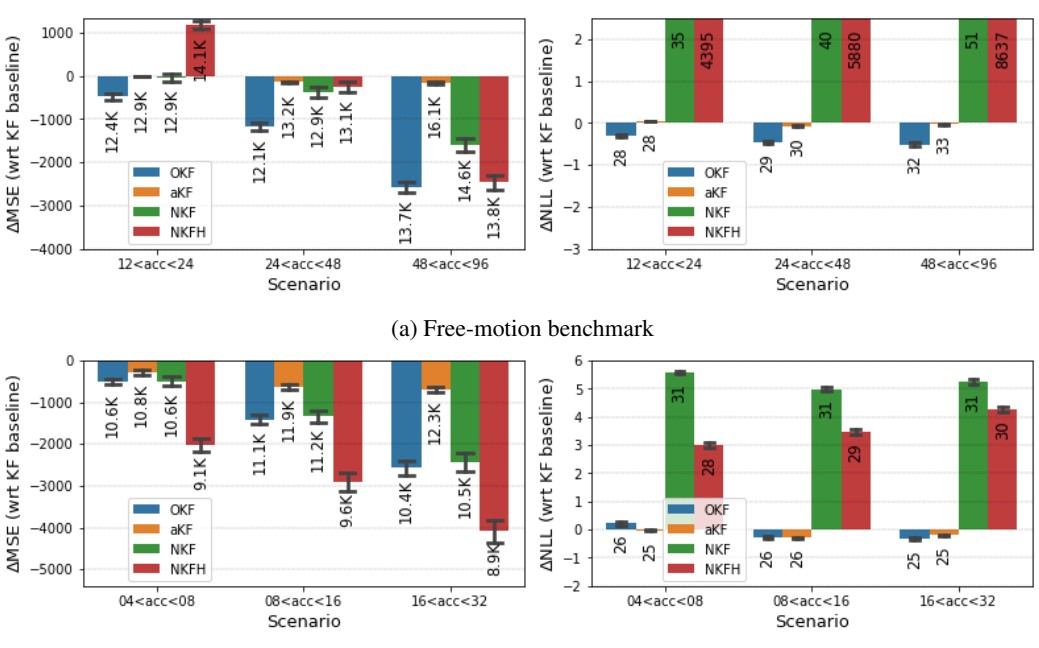

(a) Free-motion benchmark

(b) Const_a benchmark (no turns)

Figure 13: The *relative* MSE and NLL results of various models in comparison to the standard KF model. The textual labels specify the *absolute* MSE and NLL. Note that certain bars of NLL are of entirely different scale and thus are cropped in the figure (their values can be seen in the labels). In each benchmark, the models were trained with relation to MSE loss, on train data of the middle acceleration-range: the two other acceleration ranges in each benchmark correspond to generalization over distributional shifts.

Figure 13a shows that in the free-motion benchmark, all the 3 neural models improve the MSE in comparison to a standard KF, but lose to an optimized one (OKF). Furthermore, while OKF has the best NLL scores, the more complicated models NKF and NKFH increase the NLL in orders of magnitude. This issue may be handled by explicitly optimize the NLL in addition to the MSE, but such multi-loss optimization is out of the scope of this work. Note that the instability of NKFH is expressed in poor generalization to lower accelerations in addition to the extremely high NLL score.

Figure 13a shows that in Const_a benchmark, all the 3 neural models improve the MSE in comparison to a standard KF, but only NKFH improves in comparison to OKF as well. Note that while NKFH does better in this case than in the free-motion benchmark, it still suffers from very high NLL.

In summary, by comparing the optimized neural models to the standard KF model, all the 3 models would be found superior to the KF in both benchmarks in terms of MSE. However, when shifting the baseline to an *optimized* KF, we find that the neural aKF and NKF are in fact inferior to OKF, and that the comparison between NKFH and OKF depends on the selected benchmark and metric.

Of course, our results do not imply that neural-networks in general cannot be superior to a KF: only that when comparing the two, one must optimize the KF similarly to the neural model to avoid misleading experimental results. As discussed in Section 6, the wrong methodology of using a non-optimized KF baseline is very common in the literature.

## L    ON THE GUARANTEES OF GRADIENT-BASED OPTIMIZATION

Method 2 can rely on any gradient-based optimization algorithm, such as SGD or Adam (Diederik P. Kingma, 2014) (the latter is used in this work). Such optimization algorithms have achieved remarkable results in a variety of optimization problems in the recent years. This includes impressive results in non-convex problems where local-minima exist (Zhong et al., 2020), as well as generalization to unseen data (Neyshabur et al., 2017).

Compared to the problems where algorithms like Adam are often applied (e.g., language models with more than 100M parameters (Devlin et al., 2019)), typical filtering problems such as the ones experimented in this work are arguably small and simple to optimize. Thus, the robust results of Method 2 – namely, the consistent advantage over Method 1 in *all* the experiments in this work – should not come as a surprise.

Having said that, the *theoretical* guarantees of the gradient-based algorithms are limited for non-convex problems. Note that the dependence of our loss function (filtering errors) on the model parameters (after the Cholesky parameterization) is indeed not convex. In such settings, convergence to a local optimum of the loss *is* in fact guaranteed: Adam, for example, is guaranteed to converge given a twice-differentiable loss function (Bock and Weiß, 2019). Stronger convergence guarantees exist for the SGD algorithm under certain conditions (Fehrman et al., 2020), yet in general, gradient-based optimization algorithms do not necessarily converge to the global optimum of the loss.

Note that in all the problems experimented in this work, the standard Method 1 (noise-estimation) has no optimality guarantee either, as the KF assumptions do not hold. Indeed, the experiments demonstrate its sub-optimality, and Appendices E,F prove it theoretically for a sample of problems.

## M    DETAILED RESULTS

Figures 14-17 below provide additional details about the experiments discussed in Sections 5,4.

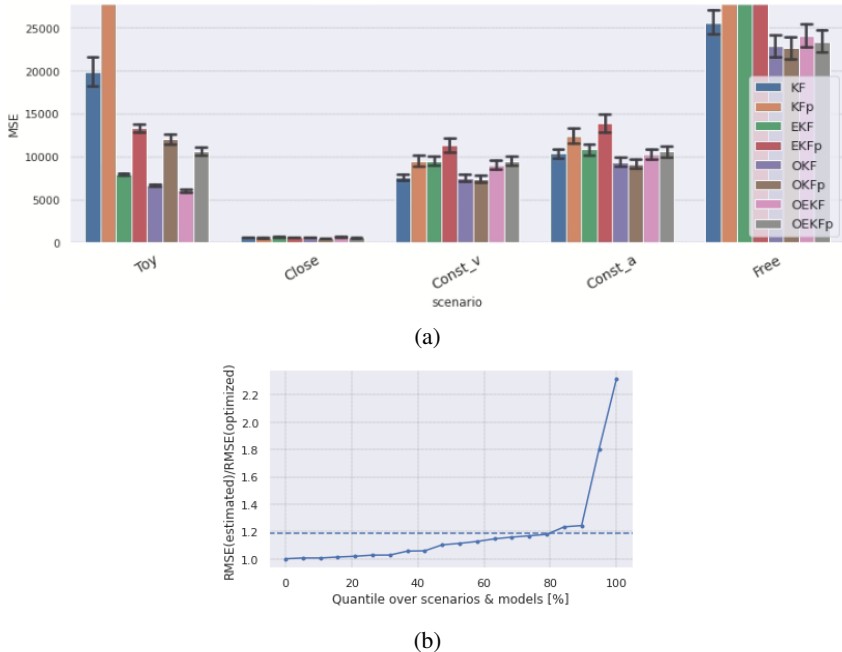

(a)

(b)

Figure 14: (a) Errors of different tracking models over different benchmarks. This is a different visualization of the results of Table 1. The error-bars correspond to confidence intervals of 95%. Note that the optimized models (the last four) tend to yield better and more stable results than the estimated models (the first four). (b) The RMSE ratio between the estimated models and the optimized ones, over all the benchmarks and designs discussed in Section 4. Note that all the ratios are above 1, i.e., all the models in all the benchmarks had smaller errors when tuned by optimization. The horizontal line represents the average ratio.

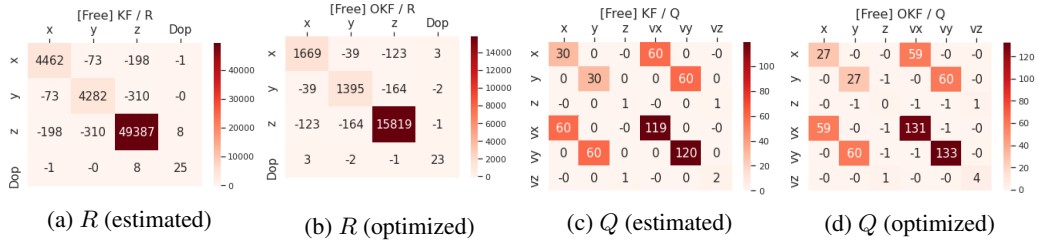

(a) $R$ (estimated)          (b) $R$ (optimized)          (c) $Q$ (estimated)          (d) $Q$ (optimized)

Figure 15: The observation noise ($R$) and prediction noise ($Q$) matrices obtained in a (Cartesian) KF by noise estimation and by optimization with relation to $MSE$, based on the dataset of the Free-motion benchmark.

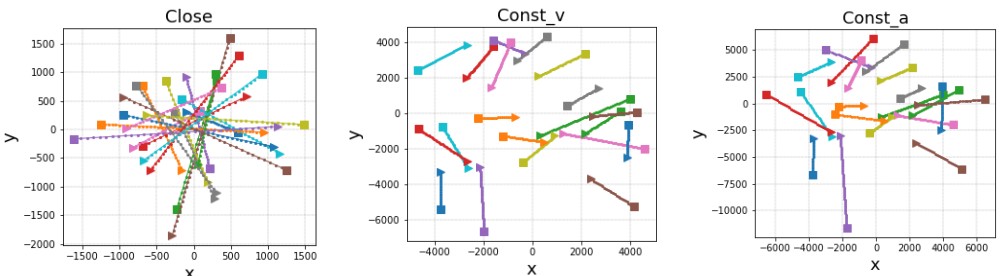

Figure 16: Samples of targets trajectories in the various benchmarks described in Section 4, projected onto the horizontal plane. This is an extension of Figures 2b,2c for the rest of the benchmarks.

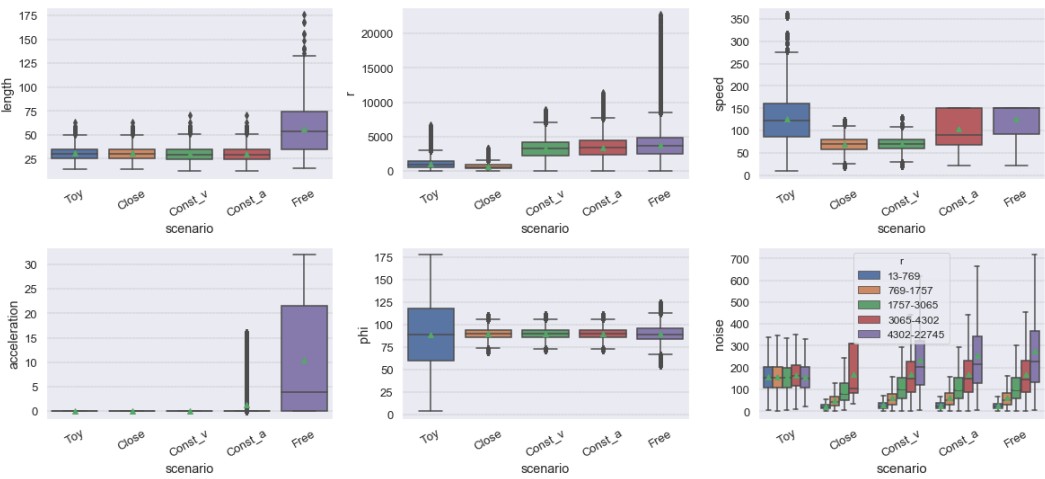

Figure 17: Descriptive statistics of the various benchmarks of Section 4: duration of targets trajectories; distance from the radar; targets speed; targets acceleration; motion direction (90 degrees correspond to horizontal motion); and observation errors vs. distance from radar.

