# OpenReview forum: "Kalman Filter Is All You Need: Optimization Works When Noise Estimation Fails"
_ICLR.cc/2022/Conference — ICLR 2022 Submitted_

### Official Review · Reviewer_D5C1 · 2021-11-02

**Correctness:** 3
**Technical Novelty And Significance:** 1
**Empirical Novelty And Significance:** 2
**Recommendation:** 3
**Confidence:** 3

**Main Review:**

Strengths

+ The problem of estimating state from noisy observations for nonlinear systems is an open and important challenge, and the proposed method is intuitive and simple to implement, and appears to yield good empirical performance.

+ The paper is clearly written and easy to follow.

+ The empirical evaluations are comprehensive, in that many scenarios and baselines are considered.

Weaknesses

- Some of the more minor contributions are in fact well established.  While I completely agree that highlighting the shortcomings of the Kalman Filter in an expository manner provides excellent motivation for this work, the idea that state-estimation was a solved problem since Kalman and that the paper is reopening this solved problem is incorrect and a bit over the top.  That the Kalman Filter is only optimal under strong assumptions, and that it can fail spectacularly when these assumptions are violated, is very well known and hardly a contribution. Similarly, using a Cholesky parameterization for optimizing over PSD matrices is standard.  In light of this, I would recommend that the grandiose tone taken in the introduction perhaps be pared back a little bit, as I found it distracted from the otherwise nice insights of the paper.

- Definition 2 is formulated under the assumption of a linear-time-invariant system; however, Kalman Filtering approaches in the time-varying setting also exist, see for example the textbook "Linear Estimation" by Sayed, Hassibi and Kailath.  That no such time-varying approach was used as a baseline was also disappointing, as this could likely have compensated for some of the errors introduced by nonlinear dynamics.

- There has been a recent line of work on regret minimization based Kalman Filtering, which updates parameters/estimates online to compensate for the fact that "effective noise" is in fact rarely Gaussian or as modeled.  That no comparison to these baselines is presented is also disappointing.  These methods are further completely online, and require no labeled datasets for supervised learning.  Relevant papers to look at include:

@article{goel2021regret,
  title={Regret-optimal estimation and control},
  author={Goel, Gautam and Hassibi, Babak},
  journal={arXiv preprint arXiv:2106.12097},
  year={2021}
}

@article{tsiamis2020online,
  title={Online learning of the kalman filter with logarithmic regret},
  author={Tsiamis, Anastasios and Pappas, George},
  journal={arXiv preprint arXiv:2002.05141},
  year={2020}
}


**Summary Of The Paper:**

This paper proposes using gradient-based optimization to tune the state and noise covariance parameters defining a Kalman Filter via supervised learning (i.e., assuming access to ground truth state measurements during training).  The need for this approach is motivated by the stringent assumptions under which optimality of the Kalman Filter is shown, and that these assumptions often fail in practice.  It is shown through several case-studies that the Optimized Kalman Filter (OKF) not only significantly outperforms a baseline KF implemented with estimated noise covariances, but also matches or outperforms Extended Kalman Filters and an LSTM-based "neural KF" introduced by the authors.

**Summary Of The Review:**

While the paper highlights and addresses an important problem, and proposes a simple yet effective solution, I believe that without a comparison to *online Kalman Filtering* approaches that minimize regret, it is impossible to determine if this truly represents an improvement over the state of the art in terms of empirical performance.  From a theoretical perspective, I see little to no novelty in the paper's contributions.

---

> ### Author Response · Authors · 2021-11-11
> **Contribution claims rephrased for clarity; baselines suggested by the reviewer do not address the same problem**
>
> We thank the reviewer for the comments, and look forward to a fruitful discussion. Rereading our claims in the introduction, we see how some of them could be interpreted differently than intended, and we rephrased accordingly (in particular in the contribution paragraph). Regarding the baselines for comparison, if we understand the suggestions correctly, they correspond to inherently different frameworks and cannot address the problem presented in the paper, as explained below. We hope that the reviewer can clarify if we misunderstood any of their comments, or otherwise reconsider their recommendation.
> ___
> 1. **Novelty and correctness of claims**:
> * “the idea that state-estimation was a solved problem… is incorrect” - this is indeed not our claim, and we cite many counter examples. Our claim is that *tuning KF parameters* given *ground-truth training data* was considered solved - the gold-standard is to calculate the sample covariance matrix of the noise. Having said that, we are willing to tone down the claim about reopening a closed problem, if after the rephrasing the reviewer still sees it as a source for unclarity.
> * We indeed cite the Cholesky parameterization as an existing technique for SPD optimization. We now also refined the paragraph discussing it in Section 3.
> * “that the KF is only optimal under strong assumptions… is well known” - the actual claim is that the *parameters tuning* of KF is only optimal under the assumptions. And while no user would explicitly claim otherwise, **the sub-optimality still often goes unnoticed** - not out of ignorance, but due to sensitivity to details. An excellent example is our current discussion with Reviewer Eeik, where we try to agree on which analytical model is optimal in the Toy benchmark. **Thus, we believe that showing exactly how violating different assumptions causes different changes in the optimal parameters is an important contribution**.
> * The contribution discussed above justifies the need for **our main novelty: a simple method for tuning the KF from ground-truth data, which is robust to (noticed and unnoticed) model misspecifications**. This novelty does not pretend to be complicated - just putting together a few concepts to obtain lower errors in an important set of problems, in a way that is not currently done.
> * Another important contribution is deducing and demonstrating the popular erroneous methodology, where optimized complicated learning models are compared to a non-optimized KF.
> 2. **Time-varying model**:
>
> The problems discussed in the paper have stationary dynamics: the dynamics may be state-dependent (e.g. H=H(x_t)) but are not time-dependent. Thus, there is not a clear way to choose time-dependent models H_t or F_t, and no reason to believe they will help.
>
> Regarding the dependence on (the time through) the state - note that in the Doppler radar problem, where indeed H=H(x_t), our baseline KF implementation *does* exploit this dependence by using H=H(z_t) (as an approximation for the unknown H(x_t)). This is a common practice, but it was not stated clearly in the text before, and we now clarify it in Section 4.1 and Appendix A.3.
>
> The distinction between time-invariant and time-dependent dynamics relates to the setting and is orthogonal to parameters optimization: both the standard noise-estimation and our optimization can be trivially generalized to a setting with given F_t & H_t instead of F & H.
>
> 3. **Online regret minimization**:
>
> While the motivation is similar, the online framework is quite different: Tsiamis and Pappas, for example, rely on *the current trajectory*, with *no ground-truth data* (hidden-states), to predict the next *observation*. We rely on *offline data*, with *available ground-truth*, to predict the *hidden-state*. The online setting is thus substantially different and while interesting on its own, not the topic of our work.
>
> Supervised learning from an offline dataset is a common practice in filtering problems (see Section 6), and by no means is a special case of online learning. In such a setting, online learning (1) does not exploit most of the data, and (2) requires updates in runtime - which is often inapplicable to latency/compute-sensitive systems. If you need to monitor the plane that has just taken off, and you have a dataset of 100 trajectories X 30 time steps X [observation + hidden state], you don’t try to build a new model from the 5 radar measurements collected in the last few seconds.
>
> That being said, the distinction of our framework from online regret-minimization is important and we now added it to the related work section. Thanks for pointing this out.
>
> Finally, it is worth noting that we take a prominent algorithm used in every plane, car, and cell phone, and make it work better in a significant number of applications, just by changing its parameters. The possibility of addressing some of the shortcomings of the Kalman filter with online learning should not detract from our contribution.

---

### Official Review · Reviewer_Qzvj · 2021-11-03

**Correctness:** 3
**Technical Novelty And Significance:** 2
**Empirical Novelty And Significance:** 2
**Recommendation:** 6
**Confidence:** 4

**Main Review:**

(+) The paper is well-organized and well-written.

(+) Theoretical analyses of the proposed optimization method.

(-) Noise estimation in KF models is a classical and open issue. The authors should carefully claim their contributions on this point.

(-) The proposed method is a supervised learning method - ground truths are required - this is the major downside as compared to the regular KF methods.

(-) Cholesky decomposition-based gradient descent is not novel for SPD matrices estimation.

(-) Only benchmark datasets were used to demonstrate the superiority to the baseline KF. The data from practical applications are expected. Although a few samples of the filtering results over the real-world data were given in the appendices, no statistical comparisons were shown.

(-) It is not clear on the explanation of the benchmark results - what is the scale of the errors obtained? They should be relative or percentage values, to better understand the improvements.

(-) As ground truths are involved, it is interesting to see how the machine learning methods other than KF-related perform. It is reasonable to compare the proposed approach to the KF-related methods only. However, the ground truths are enforced, which are the most costly and most difficult to obtain in practice. From this perspective, it would be necessary to demonstrate the proposed KF approach performs comparably against these supervised machine learning methods other than KF-related.

**Summary Of The Paper:**

This paper targets the design of a classical filtering method - the Kalman filter (KF). The linearity assumption is a strong limitation of KF models although a wide range of variants have been demonstrated for non-linear systems. Different from these studies, the authors focused on the estimation of the noise models in the KF(-class) models, in order to improve the accuracy and robustness of the KF estimates, through an optimization method. The proposed approach was assessed on a benchmark dataset, and the results demonstrated the superiority of the proposed method. Also theoretical analyses are provided in the appendices.

**Summary Of The Review:**

Please refer to the weaknesses in the above section.

---

> ### Author Response · Authors · 2021-11-11
> **Responses to comments**
>
> We thank the reviewer for the helpful comments. As discussed below, we now did some rephrasing to ensure the clarity of our claims.
>
> 1. **Re-opening a solved problem**: The problem that was considered solved is tuning of KF *given available ground-truth (hidden-states) data for training*: the gold-standard is to calculate the sample covariance matrix of the noise. We now rephrased this claim in Section 1 to ensure it is unambiguous.
>
> 2. **Scope**: We indeed focus on filtering problems with available ground-truth training data, as we point to multiple times in the paper (e.g., in the *limitations* paragraph in Section 1). This is a standard framework, as can be seen in the references in Section 6. Our method is intended to leverage the supervised data whenever it is available; other methods are intended to work without it and thus cannot exploit it. This is not a weakness or a strength, but simply the scope of the problem.
>
> 3. **Cholesky parameterization**: We essentially claim novelty only for using it to optimize the KF from ground-truth data. We now rephrased the corresponding paragraph at the end of Section 3 to make it clearer.
>
> 4. **Real-world data**: Results are shown in Figure 3 in the front paper (the video tracking task), including statistical-significance metrics.
>
> 5. **Scale of errors**: In all the experiments we report the absolute errors (not only relative ones), so the scale should be clear. Did you refer to the relative errors in the barplot in Figure 5? If so, the absolute errors are shown as labels in the figure (as explained in the caption).
>
> 6. **Comparison to supervised learning models**: this is addressed in Section 5, where we compare our method to an LSTM-based model (and to additional models in Appendix K). Note that we keep the framework of the KF to provide the whole posterior estimation of the state, but the point-estimate of the predictions is provided directly by the supervised LSTM.
>
> Finally, regarding the second (+): our theoretical analysis explains why noise-estimation is not optimal, thus justifying the necessity of optimization. However, we do not provide theoretical analysis of the proposed gradient-based optimization, which is a whole field of research by itself (see the new Appendix L).

---

### Official Review · Reviewer_NGZA · 2021-11-03

**Correctness:** 1
**Technical Novelty And Significance:** 2
**Empirical Novelty And Significance:** 2
**Recommendation:** 3
**Confidence:** 2

**Main Review:**

The main contribution of the paper is not clear to me.
It looks like that the paper tackles the problem of noise estimation of the Kalman filter in an alternative method. However, the paper is not written in a cohesive way which makes it very difficult to follow what it narrates. It is not clear how different parts of the paper are related to each other.
The paper contains many well established concepts that looks unnecessary to include in the paper.
The title of the paper does not explain what has been tackled in the paper.


**Summary Of The Paper:**

The paper claims that optimization of the Kalman filter parameters are needed in cases where the filter assumptions are violated.

**Summary Of The Review:**

The paper is not written well and does not have a flow. The rationale and objective of the paper is not clear. Therefore, I can not recommend it for publication.

---

> ### Author Response · Authors · 2021-11-11
> **Response to comments**
>
> The main claim of the reviewer is that the paper is unclear. The 3 other reviewers wrote:
> * “I think the message that the paper is trying to convey is very clear” (Eeik).
> * “The paper is well-organized and well-written” (Qzvj).
> * “The paper is clearly written and easy to follow” (D5C1).
>
> The introduction explains our goal, framework, and “how different parts of the paper are related to each other”, as put by the reviewer. If there are more concrete unclarities, we will be happy to address them and appreciate the reviewer’s help in identifying those.

---

### Official Review · Reviewer_Eeik · 2021-11-04

**Correctness:** 3
**Technical Novelty And Significance:** 2
**Empirical Novelty And Significance:** 3
**Recommendation:** 5
**Confidence:** 4

**Main Review:**

The paper is nice to read. I think the message that the paper is trying to convey is very clear, important and impactful. I appreciated the simplicity of the message, the style of the paper, and the effort in addressing its limitations and connection to related work.

However, I am not convinced with the main message that the  optimization based procedure is better than noise estimation. In particular, I am not sure how do the authors evaluate the noise estimation for nonlinear setting, since the formulas given in paper are only for linear observation model. If, for nonlinear observation model y = h(x) + w , the noise covariance is estimated with Cov[y-h(x)] from data, then this is indeed the optimal estimator for the covariance matrix in mse. For a trajectory (x_k,y_k), the difference y_k - h(x_k) = w_k, where w_k are i.i.d, and empirical covariance of w_k is the best mse estimate for cov(w). So I would appreciate if the authors explain the noise estimation for nonlinear observation model and theoretically why is it bad. I read appendix E, but I did not understand the need for defining \tilde{H}, why not use the full nonlinear observation model to estimate R.

Moreover, it is not clear if the proposed optimization procedure has a unique solution. For example, in (Formentin and Bittanti, 2014), it is shown that the KF estimate depends only on the ratio of noise covariances in scalar case. Therefore, the MSE optimization problem can only recover the ratio between noise covariances. So, it would be great if the paper have a clear and precise statement of the optimization problem it aims to solve, and analysis on why the minimizer is unique.

Also, a main disadvantage of the proposed procedure is that the optimization problem is highly nonlinear and nonconvex, specially after Cholesky decompostion. The paper does not provide any guarantee for its convergence, unlike estimation which simply follows from law of large numbers.

Some minor questions/issues:
1- I could not find the result in (Humpherys et al., 2012) that the paper refers to.

2- It will be good to add some classical adaptive filtering papers by Mehra and Carew and Bélanger.

3- I don't think KF requires F and H to be time-invariant.

4- I was confused by polar coordinates in 3d. Does it mean spherical?

5- Does the proposed procedure have any advantage for linear Gaussian setting?








**Summary Of The Paper:**

The paper studies the problem of using Kalman filter for estimating the state of a dynamical system when the noise covariance matrices, for both state dynamics and observation, are unknown. The paper assumes access to trajectories of both state and observation. In this setting, a natural approach to solve the problem is to use the data to form an estimate for the covariance matrices. However, the paper argues that an optimization procedure to find noise covariance matrices to minimize the MSE is favorable and should become the "new standard procedure for KF tuning". With several numerical experiments, the paper illustrates that the optimization based KF tuning provides much better and robust result compared to standard KF based on estimation, and the comparisons made in the advanced neural network based estimation literature is not fair.






**Summary Of The Review:**

I recommend marginally below acceptance threshold. I think the paper is well-written, but it should be much more precise mathematically, explaining the estimation procedure for nonlinear setting, and address some fundamental questions about the optimization problem.

---

> ### Author Response · Authors · 2021-11-11
> **Correctness explained; estimation definition clarified; numeric optimization discussion added**
>
> We thank the reviewer for the helpful comments. We modified the paper according to the comments, and as explained below, we believe that the current discussion only proves the necessity of our work.
>
> 1. **Correctness and noise estimation**:
> * Thanks for pointing out the unclarity. We now clarify the implementation of the KF and the tuning methods 1,2 given the non-linear Doppler model (mentioned in Section 4.1 and discussed in detail in Appendix A.3).
> * **We did in fact exactly what you said**: R:=Cov[y-h(x)] (or in our notation: Cov[z-h(x)] = Cov[z-H(x)*x]).
> * This is **not** optimal in terms of estimation-errors of x. It would be optimal if we knew h(x) in runtime. However, as we don’t know x in runtime, we can only use an approximation of h(x) (denoted \tilde{H} in Appendix E), which effectively adds noise to the filtering, leading to the results in Section 4.2.
> * There *is* in fact an optimal analytical correction in this case, discussed in Appendix E: R:=Cov[z-H(z)*x].
> * **This discussion demonstrates exactly how trying to pick the optimal analytical solution per setting is prone to errors**. And this discussion only refers to the Toy benchmark with the Cartesian KF baseline, out of tens of settings in the paper. One could look for analytical tricks in every single setting, which may be infeasible, tedious and prone to errors; or instead, one could simply use our optimizer.
>
> 2. **Gradient-based optimization**:
>
> * That’s an important discussion which is now added to the *limitations* paragraph (Section 1) and in more detail to Appendix L. Thanks!
> * Convergence is actually guaranteed under reasonable assumptions, but only to a local optimum. There is indeed no guarantee of convergence to the global optimum.
> * Having said that, algorithms such as SGD and Adam have been shown effective and robust in a wide variety of supervised problems with much higher complexity (e.g., BERT model with 340M parameters).
> * Note that the alternative - noise estimation - has no optimality guarantees either under practical assumptions, and in fact is analytically proven to be sub-optimal in a sample of private cases (Appendices E,F).
>
> 3. **Uniqueness**: There indeed may be different valid solutions that yield the same model predictions, and the model could converge to any of them. We briefly discuss the issue in Appendix E, where Q=0 and thus the values of R only matter up to a constant factor.
>
> Regarding the minor comments:
> 1. In (Humpherys et al., 2012), the assumptions are stated in Section 3 and the optimality in Section 3.2.
> 2. Done, thanks.
> 3. The KF framework is sometimes defined with F,H and sometimes with F_t,H_t. While the dynamics often depend on the unknown state X, they usually do not depend directly on the known time t, and thus we prefered to keep the simpler formulation. Note that both noise estimation and optimization can be trivially generalized to F_t,H_t.
> 4. By polar we mean spherical. We fixed it, thanks for pointing it out.
> 5. Yes, if the noise is not i.i.d (e.g. see Appendix F). If *all* the KF assumptions are known to hold, then there shouldn’t be an advantage to optimization.

---

> > ### Comment · Reviewer_Eeik · 2021-11-22
> > **Response to rebuttal**
> >
> > Thanks for taking the time to respond to my questions.
> >
> > - Why do you assume you don't know 'x'? I thought the proposed method trains on trajectories of state and observation.
> >
> > - The non-uniqueness is an important issue that should be discussed in the paper. It effects the generalization of the proposed method, in the sense that a change in dynamics or observation requires re training.  The learned noise parameters for dynamics and observation are dependent of each other and have no meaning on their own.
> >
> > - Related to assumptions for Kalman filter in Def 2.1, it is simply wrong that KF requires time-invariant dynamics [1]. Please correct that!
> >
> > [1] Kalman, R. E. (1960). A new approach to linear filtering and prediction problems.

---

> > > ### Author Response · Authors · 2021-11-23
> > > **Response**
> > >
> > > Thanks for responding.
> > > * As usual in supervised learning, we know x on training but not on testing/inference: “Our work focuses on such problems with ground-truth available for learning (but not for inference after the learning, of course)” (from the beginning of the intro). So we can’t rely on x nor on h(x) in inference. Otherwise, our prediction model would be pred(z,x)=x.
> > >
> > > * If I suspect that the environment changed and want to update the model, I need to gather data and retrain - whether using noise estimation or supervised optimization, and even if the optimal solution were always unique. Did you mean that if I knew that specifically Q changed and R didn’t, in noise estimation I wouldn’t need to recalculate R? As I need to gather new data anyway for Q, this would hardly save me anything. In fact, using parameters optimization to compensate for errors in the modeling is the essence of our method, so in this sense retraining both Q and R is an advantage. Eventually, note that if we are not informed of the change in the environment, Appendix G (referred form Section 4) shows that empirically, our method actually generalizes better than noise estimation.
> > >
> > > * It is common to introduce the time-invariant model when the dynamics are indeed time-invariant, but you're right that we should stick to the more general framework in the introduction and the methods. We fixed it, thanks!

---

> > > > ### Comment · Reviewer_Eeik · 2021-11-24
> > > > **Thanks for the clarifications.**
> > > >
> > > > Thanks for your response and clarifications.
> > > >
> > > > 1- To make sure I understand correctly, why can't you use training data to do Cov[z-h(x)]  to learn the noise covariance?
> > > >
> > > > 2- Thanks for explaining. I am not saying this as a significant disadvantage of  your proposed method. I think the non-uniqueness issue is important because the learned noise parameters, though giving the correct prediction after training, do not have physical meaning. I think this is a point a reader wants to see in the main paper.
> > > >
> > > > Regarding generalization, right now you are not considering any control input in dynamics. The proposed method suggests that the algorithm should be trained beforehand for each control input. While, a method that is based on learning the actual values of the noise parameters, can handle any control input without any retraining.
> > > >
> > > > 3- Thanks for correcting. It is true that the KF is introduced in time-invariant setting in some textbooks, but it is wrong to mention this as limitation of KF.

---

> > > > > ### Author Response · Authors · 2021-11-25
> > > > > **Thanks for the comments**
> > > > >
> > > > > 1. We do exactly that. And this is indeed the sensor noise in the physical sense - yet not the optimal parameter for state prediction!
> > > > >
> > > > > Why? As explained at the end of Section 4.2, in inference mode (testing) we must replace the unknown H(x) (e.g. by the approximation H(z)). This inserts noise to the *processing* of the observation (specifically in the Doppler component) - in addition to the noise of the observation itself - thus the *effective* Doppler noise in inference is larger. The optimization takes care of this phenomenon even if we fail to notice it; and while this whole discussion focuses on a single scenario, the optimization also handles tens of other scenarios in our experiments - each with its own (known and unknown) issues.
> > > > >
> > > > > 2. Thanks for pointing this out, we will add it to the discussion. The optimized parameters may actually reveal interesting physical meaning (e.g. as with the Doppler noise discussed above), but this meaning is indeed entangled with the other params and the state prediction process.
> > > > >
> > > > > Regarding control, this is indeed out of our scope and is not included in the problem definition. We will mention it explicitly. Optimization in this case is an important extension that may be addressed in future work.

---

### Decision · Program_Chairs · 2022-01-20

**Decision:**

Reject

**Comment:**

This paper studies the problem of estimating the trajectory of a linear dynamical system when the covariances for the process and observation noise are unknown. The standard solution is to estimate these covariances from data, and this paper instead suggests an optimization procedure. They show promising experimental results. However there are two shortcomings: In terms of theoretical guarantees, they can only show convergence to a local optimum. Moreover they assume they have access to the ground-truth hidden states. Although this is an assumption that has appeared in earlier works, it seems to limit the applicability.